# Schizophrenia, autism spectrum disorders and developmental disorders share specific disruptive coding mutations

Elliott Rees[1], Hugo D. J. Creeth[1], Hai-Gwo Hwu[2], Wei J. Chen[2], Ming Tsuang [3], Stephen J. Glatt[4], Romain Rey[1,5], George Kirov[1], James T. R. Walters [1], Peter Holmans [1], Michael J. Owen [1✉] & Michael C. O'Donovan [1✉]

People with schizophrenia are enriched for rare coding variants in genes associated with neurodevelopmental disorders, particularly autism spectrum disorders and intellectual disability. However, it is unclear if the same changes to gene function that increase risk to neurodevelopmental disorders also do so for schizophrenia. Using data from 3444 schizophrenia trios and 37,488 neurodevelopmental disorder trios, we show that within shared risk genes, de novo variants in schizophrenia and neurodevelopmental disorders are generally of the same functional category, and that specific de novo variants observed in neurodevelopmental disorders are enriched in schizophrenia ($P = 5.0 \times 10^{-6}$). The latter includes variants known to be pathogenic for syndromic disorders, suggesting that schizophrenia be included as a characteristic of those syndromes. Our findings imply that, in part, neurodevelopmental disorders and schizophrenia have shared molecular aetiology, and therefore likely overlapping pathophysiology, and support the hypothesis that at least some forms of schizophrenia lie on a continuum of neurodevelopmental disorders.

[1] MRC Centre for Neuropsychiatric Genetics and Genomics, Division of Psychological Medicine and Clinical Neurosciences, School of Medicine, Cardiff University, Cardiff, UK. [2] National Taiwan University, Taipei, Taiwan. [3] University of California, San Diego, La Jolla, CA, USA. [4] SUNY Upstate Medical University, Syracuse, NY, USA. [5] INSERM, U1028; CNRS, UMR5292; Lyon Neuroscience Research Center, Psychiatric Disorders: from Resistance to Response Team, Lyon F-69000, France. ✉email: owenmj@cardiff.ac.uk; odonovanmc@cardiff.ac.uk

Schizophrenia is a severe psychiatric disorder associated with a decreased life expectancy and marked variation in clinical presentation, course and outcome. The disorder is highly heritable and polygenic, with risk alleles distributed widely across the genome[1]. Common risk alleles collectively contribute to around a third of the genetic liability[2,3], and at least 8 rare copy number variants have been identified as risk factors[4,5]. Exome-sequencing studies have also shown a contribution to risk from ultra-rare protein-coding variants; the de novo mutation rate is modestly elevated above the expected population rate, and there is an excess of ultra-rare damaging coding variants (frequency < 0.0001 in population) in genes with evidence for strong selective constraint against protein-truncating variants (PTVs)[6–9]. *SETD1A* is currently the only gene in which rare coding variants are associated with schizophrenia at genome-wide significance[10], although another 9 genes meeting this significance threshold have been reported but not yet in a peer reviewed publication[11].

All the rare CNVs known to be associated with schizophrenia also confer risk for NDDs, that is they are pleiotropic[5]. However, these CNVs are, with one exception, multigenic and therefore it is not established that genic pleiotropy exists whereby the same genes within the CNVs increase liability to each of the disorders[12]. The only example of a single-gene schizophrenia susceptibility CNV is *NRXN1*. Consistent with genic pleiotropy, exonic deletions of *NRXN1* increase liability to schizophrenia and NDDs, but there is marked heterogeneity in the exons affected and the deletion sizes[13], leaving uncertainty as to whether precisely the same mutation can cause schizophrenia and NDD. In contrast, sequencing studies provide strong support for the hypothesis of genic pleiotropy, with genes that are enriched for ultra-rare coding variants in people with NDDs being enriched for de novo variants in people with schizophrenia[7,8]. Moreover, *SETD1A* is not only genome-wide significantly associated with schizophrenia, but also is associated with developmental disorders[14].

While *SETD1A* is enriched for PTVs in both schizophrenia and developmental disorders[10,14], previous research by us and others has been underpowered to evaluate the degree to which genic pleiotropic effects across these disorders are confined to the same functional class of variant within genes[7,8]. Moreover, although a specific PTV (c.4582-2delAG > -) in *SETD1A* has been observed multiple times in people with schizophrenia and developmental disorders[10], little is known generally about pleiotropic effects from individual rare coding variants in schizophrenia and NDDs (i.e., allelic pleiotropy). This is not a trivial point, as only allelic pleiotropy implies that equivalent changes in gene function confer risk to both schizophrenia and NDDs. Even where PTVs in the same gene lead to two different clinical outcomes, one disorder might arise due to dosage sensitivity (haploinsufficiency) whereas the other might result from PTVs that lead to truncated proteins that can have pathogenic gain-of-function or dominant-negative effects[15–17]. Furthermore, different PTVs in the same gene can affect different transcripts[18] leading to different effects. Similar considerations apply to missense variants whose functions are usually unknown, but which can have very different functional consequences for the same gene translating to different pathogenic effects[19].

Our previous study showed that de novo PTVs in schizophrenia are enriched in genes associated with an excess burden of any type of rare coding variant in NDDs[8]. However, limitations in the data that were available at that time meant we could not address two important questions. The first question asks whether in shared risk genes, the classes of mutation are the same across the disorders. The second asks whether in shared genes, the same specific mutations occur in schizophrenia and the other disorders.

These questions are key to understanding whether pathophysiologically relevant pleiotropy exists.

In the current study, we address these questions with the availability of the large and recently published NDD de novo variant data set from the Deciphering Developmental Disorders study (Fig. 1)[14]. Specifically, given that neurodevelopmental impairment is typically more severe in NDDs than in schizophrenia[20], we hypothesise that there would be a tendency for pleiotropic genes to be enriched for a more severe class of variant in NDDs than in schizophrenia. However, in contrast to our expectation, we show that genes enriched for specific classes of de novo variant in people with NDDs are also enriched for congruent variant classes in people with schizophrenia. We follow this finding with a more stringent, and conservative, test of allelic pleiotropy (Fig. 1), and demonstrate an enrichment in schizophrenia of specific variants that have been observed as de novo mutations in people with NDDs. Our findings provide strong evidence for true pleiotropic effects from rare coding variants across these disorders, thus indicating that the same changes in gene function can have very different neurodevelopmental and psychiatric outcomes.

## Results

**Genic pleiotropy.** 3438 nonsynonymous de novo variants from 3,444 schizophrenia proband-parent trios (2121 male and 1323 female probands) were obtained from 11 published studies (see Methods for clinical information about these cohorts)[6–8,21–28]. In these schizophrenia trios, de novo PTVs were significantly enriched in 127 genes associated with NDD through de novo PTVs (rate ratio = 4.89; Table 1). In contrast, no significant enrichment was observed for schizophrenia de novo missense variants in NDD PTV enriched genes (Table 1). The excess of de novo PTVs in schizophrenia was ~3.7 fold greater than that observed for missense variants in NDD PTV enriched genes (Table 1).

For 103 genes in which de novo missense variants were enriched in NDD, we observed enrichment of de novo missense variants in schizophrenia (rate ratio = 1.86) but no enrichment for de novo PTVs (Table 1). The excess of schizophrenia de novo missense variants in NDD missense enriched genes was greater than that observed for schizophrenia de novo PTVs (Table 1), but this was not statistically significant, possibly due to the small number of observations. For 53 genes that are independently associated with both de novo PTVs and missense variants in NDDs, both classes of variant were enriched in schizophrenia (Table 1), thus providing further evidence that congruent classes of variants within genes confer risk for these disorders.

To further examine the hypothesis that pleiotropic genes are enriched for congruent classes of mutation across schizophrenia and NDDs, we evaluated at a genome-wide level whether per-gene NDD test statistics for PTV and missense variants predict schizophrenia de novo variant enrichment. In the Deciphering Developmental Disorders study, independent PTV and missense enrichment statistics were both reported for 13,015 genes (missense *P*-values are not available for 6,549 genes)[14]. These data were used in a multivariable Poisson regression model to evaluate the independent effects of NDD PTV and missense gene-wide test statistics on schizophrenia de novo variant enrichment. PTV enrichment in schizophrenia was associated with the PTV enrichment test statistics in NDDs (beta = 0.15; $P = 3.3 \times 10^{-11}$) but not with the NDD missense test statistics (beta = −0.018; $P = 0.67$; Supplementary Table 1). Conversely, missense enrichment in schizophrenia was associated with the NDD missense enrichment test statistics (beta = 0.049; $P = 0.0047$) but not with the NDD PTV enrichment test statistics (beta = 0.012; $P = 0.38$;

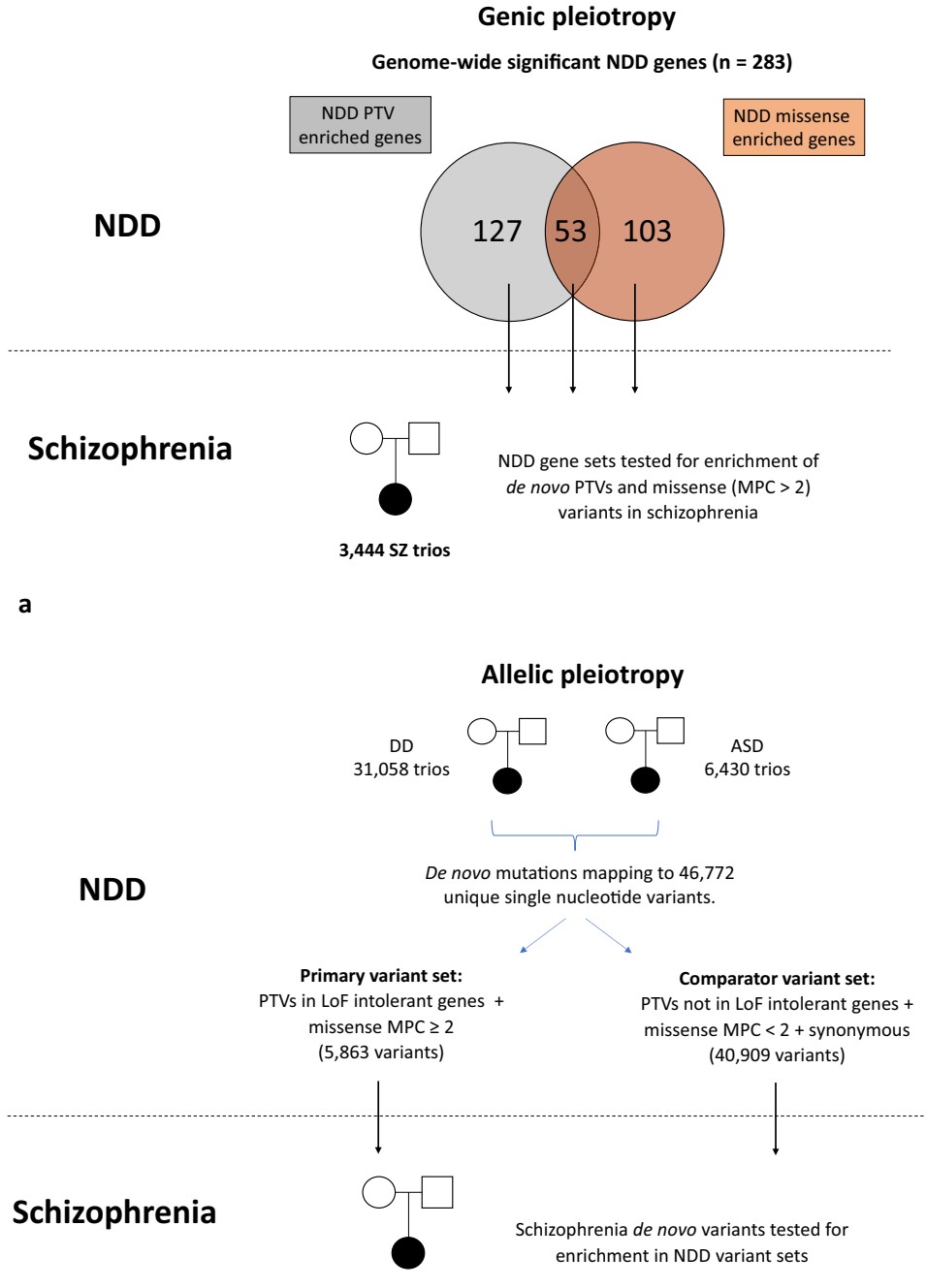

**Fig. 1 Study design for analysis of genic and allelic pleiotropy. a** Study design for analysis of genic pleiotropy. Genes enriched for protein-truncating variants or missense variants in neurodevelopmental disorders with a P value $<2.5 \times 10^{-6}$ were identified from the Deciphering Developmental Disorders study[14]. For each NDD gene set, the observed and expected number of de novo variants in 3444 schizophrenia trios was compared using a two-sided two-sample Poisson rate ratio test. See Methods for further details. **b** Study design for analysis of allelic pleiotropy. Neurodevelopmental disorder variants were defined as de novo mutations reported in the largest sequencing studies of developmental disorders[14] and autism spectrum disorders[32]. Neurodevelopmental disorder variants were stratified into a NDD primary variant set, defined as those alleles annotated as protein-truncating variants affecting loss-of-function intolerant genes or missense variants with an MPC score ≥2. The remaining neurodevelopmental disorder variants (protein-truncating variants not affecting loss-of-function intolerant genes, missense variants with an MPC score ≤2 and synonymous variants) were grouped into a NDD comparator variant set. The observed and expected number of schizophrenia de novo variants overlapping alleles within each NDD variant set was compared using a two-tailed Poisson exact test. See Methods for further details. NDD, neurodevelopmental disorder; SZ, schizophrenia; DD, developmental disorder; ASD, autism spectrum disorder; PTV, protein-truncating variant; MPC, "Missense badness, Polyphen-2, Constraint" pathogenicity score[29].

**Table 1 Enrichment of schizophrenia de novo variants in neurodevelopmental disorder (NDD) associated genes.**

| NDD gene set (N genes) | Schizophrenia de novo variant enrichment | | | |
| --- | --- | --- | --- | --- |
| | Mutation class | Obs/exp | P | Rate ratio (95% CI) |
| PTV enriched genes (127) | PTVs | 20/3.41 | $2.14 \times 10^{-8}$ | 4.89 (2.95, 7.67) |
| | Missense (MPC > 2) | 6/4.57 | 0.47 | 1.32 (0.48, 2.95) |
| | PTV vs. missense (MPC > 2) | — | 0.006 | 3.70 (1.46, 9.41) |
| Missense enriched genes (103) | PTVs | 4/2.94 | 0.79 | 1.09 (0.29, 2.80) |
| | Missense (MPC > 2) | 14/7.85 | 0.04 | 1.86 (0.99, 3.23) |
| | Missense (MPC > 2) vs. PTV | — | 0.35 | 1.72 (0.56, 5.31) |
| PTV + missense enriched genes (53) | PTVs | 7/1.64 | 0.0051 | 3.45 (1.38, 7.19) |
| | Missense (MPC > 2) | 9/3.77 | 0.014 | 2.47 (1.11, 4.83) |
| | PTV vs. missense (MPC > 2) | — | 0.52 | 1.40 (0.51, 3.82) |

Genes enriched for PTV or missense de novo variants in NDDs (significance threshold set at $P < 2.5 \times 10^{-6}$) were evaluated for enrichment of de novo variants in 3444 schizophrenia trios using a two-sample Poisson rate ratio test. *P*-values are uncorrected and two-tailed. NDD gene sets are defined as genes only associated with the given mutation class (i.e., excluding genes significant for the alternative mutation class) in the Deciphering Developmental Disorders study[14]. A Poisson regression model was used to evaluate differences between the rate of schizophrenia de novo PTVs and missense variants in NDD associated genes.

**Table 2 NDD primary variants that are observed as de novo mutations in schizophrenia.**

| Variant (chr:pos:ref:alt) | Gene symbol | NDD phenotype | Annotation | MPC score | Gene pLi |
| --- | --- | --- | --- | --- | --- |
| 12:49420670:G:A | KMT2D* | DD | PTV (stop gain) | — | 1.00 |
| 17:29679366:C:T | NF1 | DD | PTV (stop gain) | — | 0.91 |
| 7:69364416:C:T | AUTS2* | DD | PTV (stop gain) | — | 1.00 |
| X:122616672:T:C | GRIA3 | DD | Missense | 3.54 | 1.00 |
| 1:25233863:C:T | RUNX3 | DD | Missense | 2.63 | 0.84 |
| 3:11064071:C:T | SLC6A1* | DD | Missense | 2.52 | 1.00 |
| 20:472926:T:C | CSNK2A1* | DD/ASD | Missense | 2.26 | 0.99 |
| 1:173720981:C:T | KLHL20 | DD | Missense | 2.22 | 0.0094 |
| 2:166170231:G:A | SCN2A* | ASD | Missense | 2.16 | 1.00 |

'NDD phenotype' indicates whether the variant was reported in a developmental disorder (DD) trio or an autism spectrum disorder (ASD) trio[14,32]. Asterisks indicate genes enriched for de novo variants at exome-wide significance in NDDs. A full description of the NDD primary variant set can be found in the Methods. *Chr* chromosome, *pos* genomic position (in build 37), *ref* reference allele, *alt* alternative allele, *MPC* "Missense badness, Polyphen-2, Constraint" pathogenicity score[29], *pLi* probability of loss-of-function intolerance[47].

Supplementary Table 1). Our finding that genes that are pleiotropic for schizophrenia and NDDs tend to be enriched for congruent classes of variant across these disorders was robust to inclusion of measures of gene brain expression and gene constraint as covariates to the multivariable Poisson regression model (see Supplementary Table 2 and Methods for further details). These results extend the findings based only on genes that met exome-wide significance in NDDs (Table 1), and suggest that congruent variant classes within pleiotropic genes contribute to risk of schizophrenia and NDDs.

**Allelic pleiotropy.** Of 46,772 unique single-nucleotide variants observed as de novo mutations in NDD studies (defined hereafter as NDD variants), 17 were observed de novo in 3444 schizophrenia trios (variants listed in Supplementary Data 1). NDD variants were stratified into a primary variant set, which contained variants with characteristics known to be associated with NDDs, defined as PTVs in loss-of-function intolerant genes and missense variants with MPC scores ≥ 2[29], and a comparator variant set, which contained all remaining variants (see Fig. 1 and Methods for further details). Under an allelic pleiotropy model, we predicted that schizophrenia de novo variants would be more enriched among the primary variant set than the comparator variant set.

Nine schizophrenia de novo variants were observed in the NDD primary variant set (Table 2), which is significantly greater than the number expected by chance (9 observed, 1.20 expected; $P = 5.0 \times 10^{-6}$; Table 3). The enrichment in schizophrenia for de novo variants in the NDD primary variant set was significantly greater than the general enrichment in schizophrenia for the same

types of de novo mutation in constrained genes and coding sequences - i.e. PTVs in LoF intolerant genes and missense MPC ≥ 2 mutations ($P = 1.01 \times 10^{-5}$; rate ratio (95% CI) = 6.91 (3.11, 13.38); Supplementary Table 3). Therefore, the enrichment for specific NDD variants in schizophrenia is not simply a reflection of the known modest excess of constrained de novo variants in the disorder. Schizophrenia de novo variants were not enriched in the NDD comparator variant set (Table 3), which suggests that our finding is not caused by inaccuracies in estimating the expected de novo mutation rates.

We next looked at the specific classes of variant from the NDD primary variant set, and found significant enrichment in schizophrenia for both NDD PTVs ($P = 0.01$; rate ratio (95% CI) = 6.77 (1.40, 19.80)) and NDD missense variants ($P = 0.00014$; rate ratio (95% CI) = 7.90 (2.90, 17.19)) (full details presented in Supplementary Table 4).

We sought replication of association between variants in the NDD primary variant set and schizophrenia using exome sequencing data from 4070 schizophrenia cases and 5712 controls. The rate of variants from the NDD primary variant set was ~2 fold higher in schizophrenia cases than in controls ($P = 0.036$; Table 4). The rate of both missense variants and PTVs in the NDD primary variant set were increased in schizophrenia cases compared with controls, although only the later was significantly higher ($P = 0.024$; Supplementary Table 5). The rate of variants in the NDD comparator variant set did not differ between schizophrenia cases and controls (Table 4). All variants from the NDD primary variant set observed in schizophrenia cases or controls are presented in Supplementary Data 2.

**Table 3 Enrichment of neurodevelopmental disorder variants in schizophrenia.**

| NDD variant set (N variants) | Schizophrenia de novo variant enrichment | | |
|---|---|---|---|
| | Observed / expected | P | Rate ratio (95% CI) |
| Primary variant set (5863) | 9 / 1.20 | $5.0 \times 10^{-6}$ | 7.48 (3.42, 14.20) |
| Comparator variant set (40,909) | 8 / 9.86 | 0.75 | 0.81 (0.35, 1.60) |

The number of observed and expected de novo variants from the neurodevelopmental variant sets in 3444 schizophrenia trios is shown. The NDD primary variant set includes neurodevelopmental variants annotated as PTVs in genes with "probability of loss of function intolerance" (pLi) scores ≥ 0.9 or missense variants with MPC scores ≥2. The comparator variant set includes neurodevelopmental variants annotated as PTVs in genes with pLi scores < 0.9, missense variants with MPC scores <2 and all synonymous variants. Enrichment statistics were generated using a Poisson rate ratio test. P-values are uncorrected and two-tailed. PTVs protein-truncating variants, NDD neurodevelopmental disorders, CI confidence interval.

**Table 4 Schizophrenia case-control analysis of neurodevelopmental disorders variants.**

| NDD variant set (N variants) | Schizophrenia case-control analysis | | | |
|---|---|---|---|---|
| | Case rate (N variants) | Control rate (N variants) | P | Odds ratio (95% CI) |
| Primary variant set (7979) | 0.0044 (18) | 0.0023 (13) | 0.036 | 1.90 (0.94, 3.95) |
| Comparator variant set (47,630) | 0.095 (389) | 0.097 (553) | 0.42 | 1.01 (0.89, 1.15) |

Firth's penalised logistic regression models were used to evaluate the burden of NDD variants in 4,070 schizophrenia cases and 5,712 controls. As this analysis included indels, the number of variants included in the NDD primary and comparator variant sets differ to that presented in Table 3. P-values are uncorrected and one-tailed. NDD neurodevelopmental disorders, CI confidence interval.

**Phenotypic description of neurodevelopmental disorder variant carriers**. We were able to obtain additional phenotype data for each of the 9 schizophrenia probands who carried a de novo variant from the NDD primary variant set (Supplementary Data 3). Four probands had an age of onset before 18. Two had evidence for low premorbid cognitive function; one of those individuals had a KMT2D PTV, achieved normal developmental milestones, attended secondary school but left without qualifications and required supported employment, the other had an AUTS2 PTV, delayed developmental milestones and required supported schooling for learning disability. None of the of other probands were reported to have developmental delay, intellectual disability, co-morbid autism spectrum disorder (ASD) or pervasive developmental disorder.

Three of the 9 schizophrenia de novo variants in the NDD primary variant set are recorded in the ClinVar database[30] as pathogenic for Okur-Chung neurodevelopmental syndrome, Kabuki syndrome, and Neurofibromatosis type 1, which are all severe NDDs associated with intellectual disability (Supplementary Data 3). Four of the 9 schizophrenia de novo variants in the NDD primary variant set are also recorded as pathogenic or likely pathogenic in the Decipher database[31] (these include all three variants recorded as pathogenic in ClinVar). The remaining 5 variants are not present in either ClinVar or Decipher databases. Where available, the phenotypes for people in the Decipher database[31] (which includes data from the Deciphering Developmental Disorders study[14]) who carry one of these pathogenic/ likely pathogenic variants are presented in Supplementary Table 6; all had moderate or severe intellectual disability and/or developmental delay, among other phenotypes such as facial and skeletal dysmorphologies, abnormalities of skin pigmentation and abnormalities of the abdominal wall or intestine. None were reported to have a diagnosis of schizophrenia or psychosis.

Three ASD probands carried a NDD primary variant that was observed in our schizophrenia sample. Two ASD probands shared the same SCN2A missense variant (Table 2), both of whom have comorbid intellectual disability (IQ scores of 50 and 69)[32]. The schizophrenia proband who had this SCN2A missense variant developed psychotic symptoms at an early age of 14 years, but does not have a learning disability or comorbid ASD (Supplementary Data 3). The other ASD proband shared a missense variant in CSNK2A1 with a schizophrenia proband

(Table 2). This ASD proband did not have intellectual disability (IQ of 92), but showed delayed age of walking[32].

A summary of the functions and conditions associated with genes affected by variants in the NDD primary variant set that were observed as a de novo mutations in our schizophrenia sample is provided in Supplementary Data 4.

## Discussion

Previous comparisons of genetic liability based on rare CNVs and ultra-rare coding variants have shown evidence for genic pleiotropy between schizophrenia and NDDs[5,7-9]. Here, we advance that work to show that pleiotropic genes are most commonly enriched for the same functional class of variant in people with schizophrenia and NDDs (Table 1). Moreover, we show that the specific de novo variants in people with NDDs are enriched in people with schizophrenia.

Given that neurodevelopmental impairment is typically more severe in NDDs than in schizophrenia[20], we hypothesised that there would be a tendency for pleiotropic genes to be enriched for a more severe class of mutation in NDDs than in schizophrenia. In contrast to our expectation, we found that genes associated with de novo variants in NDDs were enriched for the same class of variant in schizophrenia, with the evidence particularly strong for PTVs. Conversely, there was no evidence of schizophrenia de novo variant enrichment in genes associated with a different class of variant in NDD. We next conducted a more stringent analysis of allelic pleiotropy, by testing only the same set of rare coding variants in schizophrenia that occurred in people with NDDs. Here, our findings supported our hypothesis that constrained variants observed in NDDs are enriched in schizophrenia, thus providing evidence for pleiotropy at the allelic level.

Our study suggests that the same rare variants can confer risk to a range of neurodevelopmental and psychiatric outcomes, including developmental disorders, ASD and schizophrenia, thus supporting the hypothesis that at least some fraction of schizophrenia can be conceived of as part of a continuum of NDDs[20]. We did not find support for our prediction that the type of mutation is a major factor in determining the severity of neurodevelopmental outcome, though instances of this will become apparent through large-scale sequencing studies[11]. Rather our findings suggest that clinical outcome reflects additional genetic,

environmental or stochastic factors that can modify the effects of deleterious mutations. Indeed, there is evidence that the outcome for pathogenic CNVs is influenced by common genetic variation[33–35]. The situation for rare coding variants in schizophrenia is less clear pending further studies[8], but it is likely that similar considerations will apply. Our findings also indicate that the same changes in gene function can underlie both NDDs and schizophrenia, pointing to a shared molecular aetiology and therefore likely overlapping pathophysiology.

Schizophrenia and intellectual disability co-occur more often than is expected by chance, with around 3–5% of cases of schizophrenia being co-morbid[20,36]. The present findings further support the idea that co-morbidity is due to partly shared pathophysiology. However, it is unlikely that shared pathophysiology is restricted to those with co-morbid diagnoses. Many of the schizophrenia trios, including 3 of the 9 probands with a de novo variant in the NDD primary variant set, were from studies that specifically excluded individuals with intellectual disability (Supplementary Data 5). All trios, apart from 17 trios taken from the Ambalavanan et al. study[21], had been recruited from general adult psychiatric services and all probands had received a primary DSM-IV or ICD-10 diagnosis of schizophrenia or schizoaffective disorder. Furthermore, only 2 of the 9 probands with a de novo variant in the NDD primary variant set had evidence of clinically significant premorbid intellectual impairment, consistent with findings from other studies showing damaging de novo mutations may be enriched in, but are not confined to, those with intellectual disability[6,37]. These findings, together with the absence of evidence for pre-existing autism or related disorders (Supplementary Data 3), are of clinical relevance since they suggest that patients presenting with schizophrenia in the absence of neurodevelopmental comorbidities may carry damaging mutations that are associated with more severe neurodevelopmental outcomes.

These findings do not support the suggestion that carriers of NDD mutations necessarily present with clear syndromic features, based on cognitive function or NDD comorbidity. This is clearly relevant to the question of whether we should distinguish between syndromic and non-syndromic forms of schizophrenia based on IQ deficits and other features of NDDs. Currently, schizophrenia is a clinical diagnosis, using information obtained from the case history and examination of the mental state, and there is no agreed way to confirm, reject, or subclassify the diagnosis based on pathology, laboratory measures or biomarkers of any kind, including cognitive tests or features suggestive of NDDs. The idea that we might be able to define a valid subtype or subtypes based on cognitive or other features is certainly of great interest and is one we have considered carefully. However, while we acknowledge that the situation may change as more data accrue, we do not believe that there are sufficient grounds for taking such a step. Rather, we believe that the evidence better supports the view that there is a spectrum of neurodevelopmental impairment in those with schizophrenia, rather than a dichotomy between disorders that represent "true schizophrenia" and disorders that can be considered pure manifestations of a genomic disorder or syndrome[38]. For example, IQ is continuously distributed in schizophrenia, but with a mean reduction of approximate 1 standard deviation relative to the general population[39]. The evidence also suggests that IQ has a monotonic relationship with schizophrenia risk across the IQ range[40] rather than there being an IQ threshold that is associated with a step change in liability. Moreover, lower cognitive ability in schizophrenia is not associated with a lower burden of common alleles that confer risk to what might be considered general forms of schizophrenia[41]. Finally, for individuals in whom it is very tempting to assign schizophrenia to a syndromic cause, i.e., people with deletion of 22q11.2, schizophrenia is the result of both the CNV and common variant liability that is shared with general forms of the disorder[35]. Thus, while we cannot exclude the idea that true syndromic causes of schizophrenia may be identified in the future, for now, we do not think a dichotomy based on cognitive ability or NDD features is justified. This highlights the need for additional studies to systematically evaluate the effect of NDD variants on quantitative measures of cognitive function and other biomarkers in schizophrenia.

We also observed pleiotropic effects for variants known to be pathogenic for several syndromic developmental disorders, suggesting that schizophrenia should be included among the phenotypes associated with these mutations. It is possible for features of schizophrenia to sometimes be missed in individuals diagnosed with NDDs, particularly in those with impaired communication skills. It is also the case that cognitive function is frequently not tested clinically in schizophrenia and that functionally impactful impairments, whether or not they satisfy formal diagnostic criteria for ID, are frequently missed. This argues that assessment of cognitive function should be an integral part of the clinical workup for schizophrenia as recently suggested in ICD-11[42]. Thus, our findings highlight the clinical need for greater awareness of potential comorbidity between schizophrenia and neurodevelopmental disorders.

Given the substantial enrichment over expectation of NDD variants in our primary variant set-based analysis (~7.5 fold), our analysis of allelic pleiotropy implicates variants in KMT2D, NF1, AUTS2, GRIA3, RUNX3, SLC6A1, CSNK2A1, KLHL20, and SCN2A in schizophrenia. However, definitive implication of any individual gene, much less any individual variant, requires much larger datasets than those currently available. Nevertheless, the following 4 variants are defined as pathogenic/likely pathogenic in the ClinVar[30] and/or Decipher[31] databases, which adds to the probability that they confer risk to schizophrenia: (1) The missense variant in CSNK2A1, which encodes an alpha subunit of casein kinase II, is pathogenic for Okur-Chung neurodevelopmental syndrome (OMIM #617062), an autosomal dominant disorder characterised by intellectual disability and dysmorphic facial features. (2) The stop-gain PTV in KMT2D, which encodes a histone methyltransferase that methylates the Lys-4 position of histone H3, is pathogenic for Kabuki syndrome (OMIM#147920), an autosomal dominant disorder characterised by intellectual disability and additional dysmorphic facial features. (3) The stop-gain PTV in NF1 encodes neurofibromin and is pathogenic for neurofibromatosis, type 1, an autosomal dominant condition associated with learning disabilities and tumours of nerves and skin. (4) The stop-gain PTV in AUTS2 (Activator Of Transcription And Developmental Regulator) is recorded as likely pathogenic in Decipher. A summary of the 9 genes implicated in schizophrenia by our allelic analysis is presented in Supplementary Data 4.

In conclusion, we performed a genome-wide study of allelic pleiotropy from rare coding variants for schizophrenia and NDDs, and show sets of genes associated with NDDs are enriched for congruent classes of variant in schizophrenia, as well as identify specific variants enriched for pleiotropic effects across both disorders. Collectively, our findings support the hypothesis that schizophrenia forms part of a continuum of NDDs including ASD and developmental disorders[38]. Our study points to a shared molecular aetiology and the need for more work exploring the mechanistic and clinical relationships between NDDs and schizophrenia.

## Methods
**Ethics statement**. All research conducted as part of this study was approved by the Research Ethics Committee for Wales and consistent with regulatory and ethical guidelines.

**Table 5 Summary of single-nucleotide variants included in the allelic pleiotropy de novo analysis.**

| Phenotype | N trios | Total DNVs (PTVs, miss, syn) | Unique DNVs (PTVs, miss, syn) |
|---|---|---|---|
| Schizophrenia | 3444 | 3208 (186, 2197, 285) | 3207 (186, 2197, 824) |
| DD[14] | 31,058 | 40,818 (3,638, 28,193, 8,987) | 39,560 (3400, 27,211, 8949) |
| ASD[32] | 6430 | 7337 (516, 4954, 1867) | 7306 (514, 4934, 1858) |
| NDD (ASD + DD) | 37,488 | 48,155 (4154, 33,147, 10,854) | 46,772 (3900, 32,076, 10,796) |

The 'N DNVs' column shows the total number of de novo missense, synonymous, stop-gain, splice-donor or splice-acceptor variants reported in the respective phenotype after excluding variants on the Y chromosome or in mitochondrial DNA. The Unique DNVs column shows the number of de novo variants observed in the respective phenotype after excluding duplicate variants. *DNV* de novo variant, *PTV* protein truncating variant, *miss* missense variant, *syn* synonymous variant, *DD* developmental disorder, *NDD* neurodevelopmental disorder (ASD + DD).

**Schizophrenia de novo data**. De novo variants from 3444 schizophrenia proband-parent trios (2,121 male and 1,323 female probands) were obtained from 11 published studies (Supplementary Data 5)[6–8,21–28]. The probands were ascertained from psychiatric wards or outpatient clinics, and all had received a DSM-IV (Diagnostic and Statistical Manual of Mental Disorders; fourth edition) or ICD-10 (International Statistical Classification of Diseases and Related Health Problems; 10th revision) research diagnosis of schizophrenia or schizoaffective disorder, apart from 5 probands who had a diagnosis of non-organic psychosis (details in Supplementary Data 5).

De novo variants were re-annotated using Ensemble Variant Effect Predictor (version 96)[43]. PTVs included stop-gain, frameshift, or splice donor/acceptor variants. Missense variants were annotated with their "Missense badness, Polyphen-2, constraint" (MPC) score, which is a pathogenicity metric that combines predictions of variant deleteriousness with measures of regional missense constraint[29]. We prioritised missense variants with MPC scores ≥ 2 in our analyses, as this class of variant has been shown to be enriched in ASD cases compared with controls[32].

**Genic pleiotropy**

*Neurodevelopmental disorder gene sets*. NDD associated genes were identified from the Deciphering Developmental Disorders study[14]. In that study, 180 and 156 genes were, respectively, associated with de novo PTV and missense variants at exome-wide significance (P-value < $2.5 \times 10^{-6}$). 53 genes were independently associated at this threshold with both PTVs and missense variants. We stratified the NDD associated genes into 3 independent groups—PTV specific (127 genes), missense specific (103 genes) and PTV + missense (53 genes)—and tested each group for enrichment for de novo variants in the schizophrenia probands. The genes included in these sets are provided in Supplementary Data 6. We did not include ASD associated genes in these sets as independent PTV and missense P values were not reported in the largest published ASD study[32].

*Statistics*. For 3444 schizophrenia trios, we used published gene mutation rates to estimate the number of de novo variants expected to occur under the null in the NDD gene sets[44,45]. Where possible, gene mutation rates were adjusted for sequencing coverage; the use of unadjusted per-gene mutation would overestimate the expected number of de novo variants in these trios, and produce more conservative enrichment results[8]. A two-sample Poisson rate ratio test was used to compare the enrichment of de novo variants in NDD genes, relative to the expected number, with the enrichment observed for all genes outside of the NDD gene set relative to the expected number, thereby controlling for the minimal elevation in the background schizophrenia de novo rates. Gene set enrichment tests were conducted for two mutation classes: PTVs and missense variants with MPC scores ≥ 2.

Poisson regression was used to test for differences in the degree of enrichments for schizophrenia PTV and missense de novo variants in NDD gene sets. Here, a regression was first performed for each mutation class, with the number of observed de novo variants being the outcome variable, gene set membership (e.g. NDD associated or not) a categorical predictor, and the log of the expected number of de novo variants in each gene set category the offset. The log of the rate ratio for the enrichment of NDD associated genes in PTVs relative to missense variants is then the difference in the log rate ratios for NDD genes in the two Poisson regressions (i.e., the regression coefficient for NDD gene membership). The variance of this difference is the sum of the variances of the regression coefficients, enabling confidence intervals to be generated. The square of the difference in regression coefficients divided by the sum of the variances can be compared to a $\chi^2$ distribution with one degree of freedom to give a test of significant differences in the schizophrenia enrichment of PTV and missense de novo variants in NDD associated genes. This approach allows for the background enrichment of schizophrenia de novo variants in non-NDD genes to differ between PTV and missense variants.

We also developed a multivariable Poisson regression model to evaluate the relationship between schizophrenia de novo variant enrichment and gene level P values for PTV and missense variants in NDDs simultaneously. NDD gene P values were taken from[14]. Unlike the gene set analysis, which required an arbitrary significance threshold for a gene being considered NDD associated (i.e., P < 2.5 ×

$10^{-6}$), this Poisson regression was applied to all genes. As only 5,134 genes in our analysis have the potential to be affected by a missense variant with a MPC score ≥2, to perform a genome-wide analysis of genic pleiotropy we tested all missense mutations rather than restricting to missense variants with a MPC score ≥ 2.

N SZ variants (per gene) ~ −log(DD PTV P value) + -log(DD missense *P*-value), offset(log(N SZ expected variants))

We evaluated whether per-gene levels of brain expression and constraint against PTVs or missense variants impact pleiotropy by adding them as covariates to the above multivariable Poisson regression model. Levels of gene expression in brain tissue were obtained from the BrainSeq project and defined as the level of expression in dorsolateral prefrontal cortex averaged over all available timepoints (second trimester to 85 years)[46]. For per-gene measures of constraint, we used the gnomAD observed / expected constraint score[47], which is the ratio of observed / expected number of variants in gnomAD for a given mutation class.

**Allelic pleiotropy**

*Neurodevelopmental disorder variants*. NDD variants were identified from de novo variants observed in the largest ASD and developmental disorder proband-parent sequencing studies (total NDD trios = 37,488; Table 5), which together reported a total of 48,155 single-nucleotide de novo variants, corresponding to 46,772 unique single-nucleotide variants (summarised in Table 5, full list of variants in Supplementary Data 7). We divided these variants into a primary set and a comparator set. The primary variant set contains variants with characteristics known to be associated with pathogenicity for NDDs[14,48], namely PTVs in loss-of-function intolerant genes (genes with gnomAD pLi scores ≥0.9[47]) and missense variants with MPC scores ≥2[29]. The comparator variant set contained all remaining variants (PTVs in genes with pLi scores <0.9, missense variants with MPC scores <2 and all synonymous variants), properties that do not predict NDD pathogenicity. Under an allelic pleiotropy model, we predicted that schizophrenia de novo variants would be more enriched among the NDD primary variant set than the NDD comparator variant set.

**Statistics**. Tri-nucleotide mutation rates were used to estimate the expected per-generation mutation rates for NDD variants[29]. These mutation rates were then used to derive the number of NDD variants expected to occur de novo under the null hypothesis in the 3444 schizophrenia trios. As mutation rates have not been empirically established for indels, only single-nucleotide variants were considered (Table 5).

The numbers of schizophrenia de novo variants overlapping our primary and comparator variant sets were compared to that expected under the null using a two-tailed Poisson exact test. We also used a two-sample Poisson exact test to evaluate whether the enrichment of schizophrenia de novo variants in the primary variant set was greater than the schizophrenia background de novo rate of all PTVs in LoF intolerant genes and missense variants with MPC scores ≥2. Statistics were generated using R statistical software (version 3.4.3) and the poisson.test() function.

NDD variants in our primary and comparator sets were further evaluated using a Swedish schizophrenia case-control exome sequencing data set, which consisted of 4079 cases and 5712 controls[9]. Case-control exome sequencing data were analysed using Hail (https://github.com/hailis/hail). To test for an excess burden of NDD variants in cases compared with controls, a one-tailed Firth's penalised-likelihood logistic regression model was used, correcting for the first 10 principal components derived from the sequencing data, and for the exome-wide burden of synonymous variants, sequencing platform and sex. To focus the case-control analysis on ultra-rare alleles, as those are more likely to be pathogenic, we excluded variants with an allele count >5 in gnomAD[47]. Indel variants were included in the case-control analysis.

**Reporting summary**. Further information on research design is available in the Nature Research Reporting Summary linked to this article.

## Data availability

All schizophrenia de novo variants were obtained from the following published sources outlined in Supplementary Data 5, and from the following DOIs: https://doi.org/10.1038/

s41593-019-0565-2, https://doi.org/10.1038/nature12929, https://doi.org/10.1038/ng.886, https://doi.org/10.1038/ng.2446, https://doi.org/10.1016/j.cell.2013.06.049, https://doi.org/10.1038/srep18209, https://doi.org/10.1038/ejhg.2015.218, https://doi.org/10.1371/journal.pone.0112745, https://doi.org/10.1038/mp.2014.29, https://doi.org/10.1038/s41593-019-0564-3. De novo variants from ASD trios were obtained from Satterstrom et al.[32] (https://doi.org/10.1016/j.cell.2019.12.036), and de novo variants from DD trios were obtained from Kaplanis et al.[14] (https://doi.org/10.1038/s41586-020-2832-5). DD gene level association statistics were obtained from Kaplanis et al.[14] (https://doi.org/10.1038/s41586-020-2832-5). The Swedish case control exome sequencing data set is available through dbGaP accession number phs000473.v2.p2.

## Code availability

A description of the R functions used for statistical analysis can be found in the Life Sciences Reporting Summary. R code used for this research is available at https://github.com/reeseg/SZ_NDD_pleiotropy_analysis (https://doi.org/10.5281/zenodo.5078112[49]).

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

## Acknowledgements

The work at Cardiff University was supported by Medical Research Council Centre Grant No. MR/L010305/1 (to MJO), Program Grant No. G0800509 (to MJO, MCO, JTRW, PH) and a UKRI Future Leaders Fellowship Grant Ref. MR/T018712/1 (to ER). SJG was supported by grant R01 AG 064955 from the U.S. National Institute on Aging and grant R01 MH 085521 from the U.S. National Institute of Mental Health. DECIPHER Acknowledgments: This study makes use of data generated by the DECIPHER community. A full list of centres who

contributed to the generation of the data is available from https://decipher.sanger.ac.uk and via email from decipher@sanger.ac.uk. Funding for the DECIPHER project was provided by Wellcome. We acknowledge Drs. Peter Turnpenny, Fernando Santos Simarro, Ramsay Bowden, Joanna Jarvis, Jenny Carmichael and Andrew Green for providing their consent to publish the phenotypes observed in DECIPHER patients who have a primary neurodeve-lopmental disorder variant that is also observed in a schizophrenia patient. We acknowledge that those who carried out the original analysis and collection of the DECIPHER data bear no responsibility for the further analysis or interpretation of the data. Swedish Exome Sequencing Acknowledgments: The datasets used for the analysis described in this manuscript were obtained from dbGaP at http://www.ncbi.nlm.nih.gov/gap through dbGaP accession number phs000473.v2.p2. Samples used for data analysis were provided by the Swedish Cohort Collection supported by the NIMH Grant No. R01MH077139, the Sylvan C. Herman Foundation, the Stanley Medical Research Institute and The Swedish Research Council (Grant Nos. 2009-4959 and 2011-4659). Support for the exome sequencing was provided by the NIMH Grand Opportunity Grant No. RCMH089905, the Sylvan C. Herman Foundation, a grant from the Stanley Medical Research Institute and multiple gifts to the Stanley Center for Psychiatric Research at the Broad Institute of MIT and Harvard.

## Author contributions

E.R., P.H., M.J.O. and M.C.O.D. conceived and designed the research. E.R. analysed the data. E.R., P.H., J.T.R.W., M.J.O., M.C.O.D., H.D.J.C. and R.R. contributed to the interpretation of the results. H.H., W.J.C., M.T., S.J.G., G.K., M.C.O.D. and M.J.O. led the acquisition of the clinical samples. E.R., P.H., M.J.O. and M.C.O.D. wrote the manuscript, which was read, edited and approved by all authors.

## Competing interests

J.T.R.W., M.C.O.D. and M.J.O. are supported by a collaborative research grant from Takeda Pharmaceuticals. Takeda played no part in the conception, design, implementation, or interpretation of this study. The remaining authors declare no competing interests.
