## [Peer Review File · Nature Communications]

Schizophrenia, autism spectrum disorders and developmental disorders share specific disruptive coding mutationsReviewers' Comments:

Reviewer #1:

Remarks to the Author:

Rees et al reported a study of shared de novo nucleotide variants between schizophrenia (SCZ) and neurodevelopmental disorders (NDD). The authors found that de novo variants in SCZ are enriched in NDD. Specific de novo variants in NDDs are also enriched in SCZ. They also found a handful of pathogenic variants that present in both NDD and SCZ, albeit both conditions are not always observed in each case. They suggested that the findings support pleiotropic effects of rare variants nucleotide level, and that some forms of schizophrenia should be considered NDDs. While I can appreciate that the pleiotropic effect of rare variants at nucleotide level have not been reported in SCZ, the need of such analysis is not well explained. I also think the evidence from this study is still insufficient to support their conclusions.

1. The same analysis and results on genic pleiotropy seem to have been reported in the previous study from the same authors (Rees et al 2020 Nat Neurosci).
2. As the authors pointed out, pleiotropic effects of rare CNVs have been studied and reported. Many of the involved CNVs are recurrent with the same size, impacting the same genes. It is well known that DD, ID, ASD and SCZ share genetic etiology. What is the importance of assessing such effect at the individual genic level? Similarly, GWA studies have shown shared effects of common SNPs on psychiatric disorders (including SCZ) (2013 Lancet, Nat Genet). What can we learn from knowing the same effect on de novo variants?
3. While NDD can be assessed in SCZ subjects, can the same be done to assess SCZ in NDD cases? Most of the NDD cases were assessed in early ages. Given that SCZ is generally late onset, how can they be sure that the NDD individuals with the same variants will not develop SCZ later in life?
4. Though collectively enriched in SCZ, individual de novo missense variants are not necessarily damaging or contributing to SCZ. Other than the one in CSNK2A1 that indicated as pathogenic, how can they be sure that the variants are contributing to SCZ (Table S5)? In fact, even CSNK2A1 has "Conflicting interpretations of pathogenicity" in ClinVar.
6. Most of the additional phenotypes in Table S5 only indicate if the subjects have autism or other developmental disorder. Shouldn't the assessment be more targeted towards the known function of the genes? For example, NF1 is known also for Leukemia. Have the authors assessed that in the subject with PTV in NF1? Similarly, other than autism or developmental disorders, does the individual with PTV in KMT2D show facial features of Kabuki syndrome? Same applies to other genes reported.
7. Are the de novo variants primarily responsible for schizophrenia? Are there additional rare or common variants contributing the outcome in each case? Indeed, as the authors pointed out, it is the case in rare CNVs that other rare or common variants are also contributing. For the cases with shared variants, have they excluded the presence of other high risk variants?
8. Cross-referenced with the SCZ genes found in the latest large-scale ultra-rare variant analysis (Singh et al 2020 medRxiv), the genes that shared between NDD and SCZ here don't seem to be typically found in SCZ (perhaps except GRIA3). Does it further suggest that these shared variants are not the primary risk factors for SCZ?
9. Why did the authors only analyze de novo variants? Technically the same can be done on ultra-rare variants, which provides a greater power. Indeed, the large-scale ultra-rare variant study (these authors also coauthored) found that when focusing on the SCZ genes, "the majority of schizophrenia associations reported here appear to have little or no role in DD/ID despite the enormous power of published DD/ID studies to date". This further suggests that the de novo variants reported here are not the primary factors for SCZ.

Reviewer #2:

Remarks to the Author:

The manuscript by Rees et al describes pleiotropy of disruptive mutations in neurodevelopmental and

neuropsychiatric diseases. The results are interesting, somewhat expected and are in line with the current understanding of the genetic background of these diseases. This is a key manuscript for the field that solidifies many of the earlier assumptions.

One of the strengths is that it combines data from a large number of studies. This provides a good view of the landscape. This is a top group with a deep understanding of analytical methods.

I have only one comment. The presentation of the data used and the flow of analyses could be more reader friendly. The group uses terms like "NDD gene set", "PTV/missense gene sets", "Primary set", "Primary variant set", "negative control set" etc. Mostly these refer to variants, sometimes to patient groups. The manuscript does explain these terms, but it takes quite some digging. Would maybe a flow chart that describes the components and steps of the study help the reader to grasp the message of the study? This should be considered.

This reviewer would prefer that the Abstract would also have some actual numeric results, not just the story of the study.

Reviewer #3:

This paper describes denovo variation in neurodevelopmental disorder and schizophrenia probands. In particular, this paper addresses pleiotropy between the disorders, at the gene and allelic level, finding evidence for true pleiotropy (that is, the same mutations increasing risk for both disorders). The finding is timely with recent exome results in SCZ, for example the gene TRIO, in which LOFs seems to be enriched in SCZ cases while missense variants cause NDD; perhaps not surprisingly, the TRIO pattern is the exception and not the rule.

I thought early on the paper says that it presents new NDD cohorts. I don't see those reported in the Results. Indeed, I was expecting to see some Intro/Overview in the Results, showing the number of trio samples, the number of denovos, in new and published studies, and perhaps some clinical/phenotypic information about the cohorts.

The novelty of the true pleiotropy findings may not be that great. I suspect that many neuropsychiatric geneticists who have studied denovo variation and DenovoDB may have seen shared genes and perhaps even denovo mutations between NDDs and SCZ. Still this is a quantitative and in depth characterization, and I suggest a couple of ways to make it of greater value still.

There may be some problems with the prediction of variant category specific enrichments in both disorders, depending on the gene. First, effect sizes are stronger in NDD than SCZ, and for LOFs than for missense in both disorders; therefore, there may be limited power to detect enrichments in SCZ missense denovos in Table 1 (indeed all the rate ratios are greater than 1 and likely not significantly different from those in LOF NDD genes vs missense NDD genes). Second, some proportion of predicted damaging missense variants are essentially LOFs; therefore, LOF and missense categories should show correlated effect sizes, which would make the predicted pattern less likely to be observed. I am not sure how to take these things into account in the analysis presented in Table 1; perhaps in the test statistics regression format, both LOF and missense NDD test statistics could be added to the model to assess whether both are associated with eg missense SCZ test statistics.

The test statistic regression framework is potentially a nice one for addressing this kind of question. Genes' pLI status, or eg brain expression, could be added as covariates to test whether they impact the pleiotropy. However, it may be possible to observe test statistic correlations due to mutation rate (/gene length), in the same way we see correlations among rare gene burden test statistics which we attribute to the same genes being underpowered across disorders, so we might filter for genes with some minimum number of counts (perhaps some minimum mutation rate for denovo analyses).

Perhaps some simulations could be used to show whether this may be an issue.

SUGGESTIONS (perhaps some of these are in the manuscript, but could be emphasized more, or have analyses done and move from Discussion to Results):

Given the timeliness mentioned above, that the SCHEMA paper is forthcoming and (on the preprint server and in recent meetings) presents at least one example of differential mutation categories between the disorders, it might be nice to drill down on Table 1, to show which genes seem to drive the congruence and how many and which genes may not. Are the 20 LOFs and 6 missense SCZ denovos scattered randomly across genes or concentrated in one or a few genes?

Subphenotypes relating to cognition/school performance in SCZ probands. I would be interested in more description of this for the NDD/ASD probands (suggesting to expand the last Results subsection). Do NDD probands with denovos in SCZ denovo genes have any particular symptoms relating to cognition or psychosis? Are they more or less likely to have multi-system syndromic symptomology than NDD probands with denovos in non-SCZ denovo genes? (Is it possible to assess the frequency of different symptoms across OMIM phenotypes, for example?) Could you present some tables of the frequencies of these things, if not statistical test(s), in addition to the anecdotal list and supp table?

MINOR COMMENTS/QUESTIONS:

Table 2, are there really genes with Exac pLI $\sim 10^{-16}$? I thought those were rank percentiles across genes (ie minimum 1/20,000). Just pointing it out in case of typo.

When you look in SCZ case/control exomes, why not refer to the recent SCHEMA results (which are publicly available)?

REVIEWER COMMENTS

Reviewer #1 (Remarks to the Author):

1. The same analysis and results on genic pleiotropy seem to have been reported in the previous study from the same authors (Rees et al 2020 Nat Neurosci).

The present study is very different from the one cited by the reviewer, and we hope this is now clearer with the revisions we have introduced. Our previous study (Rees et al. 2020) showed that *de novo* protein truncating variants (PTVs) in schizophrenia are enriched among genes with an excess burden of any type of rare coding variant in neurodevelopmental disorders (NDDs). However, limitations in the data that were available at that time meant we could not address two important questions. The first question asks whether in shared risk genes, the classes of mutation are the same across the disorders. The second asks whether in shared risk genes, the same specific mutations occur in schizophrenia and the other disorders. These questions are key to understanding whether pathophysiologically relevant pleiotropy exists, and can now be addressed with the availability of the large Deciphering Developmental Disorders mutation dataset (Kaplanis et al. 2020).

We have added text to the introduction to clarify the novel aspects of our analysis of genic pleiotropy.

Page 4 paragraph 2:

“While *SETD1A* is enriched for PTVs in both schizophrenia and DD (Singh et al. 2016; Kaplanis et al. 2020), previous research by us and others has been underpowered to evaluate the degree to which genic pleiotropic effects across these disorders are confined to the same functional class of variant within genes (Howrigan et al. 2020; Rees et al. 2020)”

Page 5 paragraph 2:

“Our previous study showed that *de novo* PTVs in schizophrenia are enriched in genes associated with an excess burden of any type of rare coding variant in NDDs (Rees et al. 2020). However, limitations in the data that were available at that time meant we could not address two important questions. The first question asks whether in shared risk genes, the classes of mutation are the same across the disorders. The second asks whether in shared genes, the same specific mutations occur in schizophrenia and NDDs. These questions are key to understanding whether pathophysiologically relevant pleiotropy exists. In the current study, we can now address these questions with the availability of the large and recently published NDD *de novo* variant data set from the Deciphering Developmental Disorders study (Kaplanis et al. 2020).”

2. As the authors pointed out, pleiotropic effects of rare CNVs have been studied and reported. Many of the involved CNVs are recurrent with the same size, impacting the same genes. It is well known that DD, ID, ASD and SCZ share genetic etiology. What is the importance of assessing such effect at the individual genic level? Similarly, GWA studies have shown shared effects of common SNPs on psychiatric disorders (including SCZ) (2013 Lancet, Nat

Genet). What can we learn from knowing the same effect on de novo variants?

Pleiotropic effects have indeed been demonstrated across schizophrenia and neurodevelopmental disorders. However, as discussed in the introduction, with the exception of *NRXN1* CNVs, all CNVs that are known to increase liability to schizophrenia contain multiple genes that might credibly contribute to schizophrenia and/or NDDs, and it cannot be assumed the genes that are involved are the same. Moreover, the fact that CNVs are frequently the same size in NDDs and schizophrenia is a function of their location relative to the low copy repeats which drive recurrent mutation, it does not imply the same genes are involved in both sets of disorders. Regarding GWAS studies, very few causal variants are known, and to the best of our knowledge, there are no causal variants derived from GWAS that are known to be shared between schizophrenia and NDDs. The precision afforded by studying rare coding sequence variants in the exome allows us to show that in people with schizophrenia, pleiotropic genes have an increased burden of the same functional classes of mutation that confers liability to NDDs, and also they have an increased burden of precisely the same mutation. This implies that similar changes in gene function can confer risk to both schizophrenia and NDDs, suggesting the two sets of disorders at least partly share disease mechanisms.

The demonstration of pleiotropic rare coding variants is also of practical importance clinically, particularly for genetic counsellors. It is also important for neuroscientists who are developing animal or iPSC models of schizophrenia or NDDs, as it may dictate how functional readouts from those model systems should be interpreted in the context of human phenotypes. It also provides valuable information regarding mutations for modelling comorbidity between schizophrenia and NDDs.

3. While NDD can be assessed in SCZ subjects, can the same be done to assess SCZ in NDD cases? Most of the NDD cases were assessed in early ages. Given that SCZ is generally late onset, how can they be sure that the NDD individuals with the same variants will not develop SCZ later in life?

We are unclear exactly what the issue is here. If the concern is that the people who have these mutations only develop schizophrenia because they first had NDD, then we know this is not true as only a minority of the schizophrenia cases carrying these mutations have NDD (Supplementary Table S8). However, even if all the schizophrenia carriers had a history of NDD, it would still imply that the mutations can manifest as both NDD and schizophrenia.

It is possible that the reviewer is concerned that the shared mutations have nothing to do with the neurodevelopmental disorders in the Deciphering Developmental Disorders (DDD) sample, but are instead schizophrenia mutations that happen to be observed in the large DDD sample in the subset of people who in the future will develop schizophrenia. If that were true, our observation of these mutations in schizophrenia cases would not imply pleiotropy. However, this is not a plausible explanation for our findings. First, of the 9 overlapping alleles, 5 are in genome-wide significant DDD genes (Kaplanis et al. 2020), adding confidence that those alleles contribute to the NDDs. Second, while the size of the DDD study implies some of the participants will develop schizophrenia, our finding of shared alleles cannot be driven by this. In the absence of shared risk alleles, the lifetime risk of (future)

schizophrenia in the NDD sample is equivalent to that in the general population, that is 7/1000. Thus, in the NDD sample (37,488 trios), we expect around 262 individuals will develop schizophrenia. Given that within our much larger sample of 3,444 schizophrenia trios, we see no missense or PTV single-nucleotide *de novo* variants that occur in two or more individuals, it is highly unlikely that we would now see 9 alleles that are recurrent between our test schizophrenia sample and the much smaller sample of 262 individuals from the NDD study. We quantify how unlikely this is below.

Given no recurrent *de novo* variants in our sample of schizophrenia cases, the upper boundary for the 95% CI for the probability that a pair of schizophrenia cases share a *de novo* variant can be calculated as follows. If there are M schizophrenia cases, then there are $M(M-1)/2$ independent pairs of cases. If the probability that a pair of schizophrenia cases share a *de novo* variant in common = p , then the number of such recurrences is distributed as a binomial ($M(M-1)/2, p$). Since no recurrences were observed in our sample, the usual normal approximation for the distribution of p will be invalid, and we instead use the likelihood ratio interval (see (Brown, Cai, and DasGupta 2002)). This gives the upper 95% bound for $p = 1 - e^{(-3.841/M(M-1))}$, where 3.841 is the 95% value of a chi-sq on 1df. Thus, where $M = 3,444$ (the number of schizophrenia probands in our current study), the upper 95% CI estimate of the probability that any two probands will share a *de novo* variant is 3.24×10^{-7} . If we now introduce a new sample of N schizophrenia probands to be matched to our original sample (M), there are MN pairs of cases and the number of recurrent *de novo* variants is distributed as a binomial (MN, p). Substituting the upper 95% value of 3.24×10^{-7} for p , the probability of observing at least 9 *de novo* variants that overlap between 3,444 schizophrenia trios (M) and 262 individuals in the NDD sample is 3.29×10^{-11} . Thus, the observed overlap in *de novo* variants between schizophrenia and NDD cases cannot be due to these variants simply being schizophrenia variants.

Moreover, our observation of 9 recurrent *de novo* variants is still unlikely if the aetiologies of schizophrenia and NDD are completely unrelated, but through some ascertainment bias, the NDD samples have oversampled people such that there is a 15-fold increase in the rate of (future) schizophrenia (e.g. $N = 4000$ individuals with schizophrenia in the NDD sample; see Response Table R1 below). We note that as these probabilities are derived from the upper 95% CI for *de novo* recurrence, the true probability of observing 9 overlapping *de novo* variants may be orders of magnitude less, and accordingly the probability of making our observation of 9 recurrences under the null hypothesis considerably lower than estimated here.

N individuals with schizophrenia in NDD sample	N matching DNVs	Probability
262	9	3.29×10^{-11}
1000	9	2.71×10^{-6}
2000	9	5.17×10^{-4}
3000	9	7.52×10^{-3}
4000	9	0.0385
5000	9	0.112

Response Table R1. Probability of observing 9 overlapping *de novo* mutations between our schizophrenia sample (n = 3,444 probands) and individuals in the NDD sample who later develop schizophrenia. We estimate this probability for an increasing number of individuals who hypothetically later develop schizophrenia in the NDD sample.

4. Though collectively enriched in SCZ, individual *de novo* missense variants are not necessarily damaging or contributing to SCZ. Other than the one in *CSNK2A1* that indicated as pathogenic, how can they be sure that the variants are contributing to SCZ (Table S5)? In fact, even *CSNK2A1* has “Conflicting interpretations of pathogenicity” in ClinVar.

Before addressing these comments, we would like to emphasise that our conclusions rely on burden tests, they do not depend on identifying specific individual alleles that definitively contribute to schizophrenia. Nevertheless, the substantial (7-fold) enrichment for *de novo* NDD alleles in schizophrenia cases implies that the vast majority, and possibly all, of the NDD alleles we observe in schizophrenia do contribute to risk for schizophrenia. This striking enrichment is actually similar to that seen for rare coding variants in risk genes identified in the large SCHEMA schizophrenia study (OR: 3-50) (Singh et al. 2020) and in the NDD exome sequencing studies which similarly have inferred that the majority of observed alleles are pathogenic (Kaplanis et al. 2020). However, as we noted in the main text (page 17 paragraph 2), definitive implication of specific alleles will rely on sequencing studies that are much larger than those currently available for schizophrenia or NDDs.

Regarding *CSNK2A1*, the specific *CSNK2A1* missense variant we observe in schizophrenia is a recurrent *de novo* variant in unrelated individuals with Okur-Chung neurodevelopmental disorder syndrome (Chiu et al. 2018). Moreover, this specific variant is one of the most recurrent *de novo* mutations reported in the latest Deciphering Developmental Disorders study (Kaplanis et al. 2020), where it was observed in 10 NDD samples. Although this mutation is recurrent among people with NDDs, it is not observed even once among 114,704 non-neurological samples in the gnomAD data. This provides strong evidence for its pathogenic effects in NDDs. In ClinVar, there are 9 submissions with interpretations and evidence for this variant’s pathogenicity (<https://www.ncbi.nlm.nih.gov/clinvar/variation/VCV000224790.9>); 7 records classify this variant as “Pathogenic”, one as “Likely pathogenic” and one as “Uncertain significance”. Although the majority of records classify this variant as pathogenic, it has automatically been given a ClinVar Review status as “Conflicting

interpretations of pathogenicity” because of one record that was assigned “Uncertain significance”.

5. Point 5 is absent from the reviewer’s comments.

6. Most of the additional phenotypes in Table S8 only indicate if the subjects have autism or other developmental disorder. Shouldn’t the assessment be more targeted towards the known function of the genes? For example, NF1 is known also for Leukemia. Have the authors assessed that in the subject with PTV in NF1? Similarly, other than autism or developmental disorders, does the individual with PTV in KMT2D show facial features of Kabuki syndrome? Same applies to other genes reported.

The reviewer makes a good point that mutations that have large effects on neurodevelopmental disorders can have effects on non-CNS tissues and diseases. Were we doing this study prospectively, we agree it would be an excellent idea to target phenotypes linked to gene function, but this is not a prospective study. Moreover, the samples we have direct access to were recruited blind to genotype many years ago, and we do not have permission for recontact. We therefore have to rely on medical records and interviews. These include statements of characteristics that were at the time thought to be important in schizophrenia, including history of neurodevelopmental disorders, educational attainment, occupation and social functioning. Where they were present, major medical conditions were noted as well (e.g. leukaemia) but there was no systematic recording of the absence of medical comorbidities of syndromes.

7. Are the de novo variants primarily responsible for schizophrenia? Are there additional rare or common variants contributing the outcome in each case? Indeed, as the authors pointed out, it is the case in rare CNVs that other rare or common variants are also contributing. For the cases with shared variants, have they excluded the presence of other high risk variants?

Our assumption is that additional genetic and/or environmental contributions to schizophrenia act in *de novo* mutation carriers, although our conclusions remain valid regardless of whether that assumption is true or false.

We are not sure what is meant by “primarily responsible” in this context, but we take it to mean: “do they confer large effects?”. The enrichment for NDD mutations in our cases is substantial, over 7-fold, implying that most of the alleles have effect sizes at least in the range observed for CNVs and the exonic variants in genes associated with schizophrenia in the SCHEMA study. However, if “primarily responsible” is taken to mean they are the sole reason for why a person has the disorder, we would assume this would not be true. The number of cases with relevant *de novo* variants is too small for common variant burden analysis, but our own work with CNVs to which the reviewer refers (Tansey et al. 2016) informs that assumption, which is also supported by strong evidence that common variation plays a role in carriers of pathogenic *de novo* mutation with diagnoses of DD or ASD (Weiner et al. 2017; Niemi et al. 2018).

Regarding other high-risk variants, only a trivial fraction of the exome-wide rare variant contribution to schizophrenia is known, and the non-exonic rare variant contribution is completely unknown. This means we cannot exclude second rare variant hits. However, for the reviewer's information, for the 9 schizophrenia cases that carry NDD alleles, we have CNV data for only 3 of them, none of whom have a pathogenic CNV. We have access to full exome data for the same 3 cases, and none have inherited mutations in SCHEMA genes. Finally, none of the 9 cases has a second coding *de novo* variant in SCHEMA genes.

8. Cross-referenced with the SCZ genes found in the latest large-scale ultra-rare variant analysis (Singh et al 2020 medRxiv), the genes that shared between NDD and SCZ here don't seem to be typically found in SCZ (perhaps except GRIA3). Does it further suggest that these shared variants are not the primary risk factors for SCZ?

We do not think genes reported by SCHEMA should be considered primary risk factors in the sense of being the sole causes of disorder in the carriers. They also should not be considered as typically found in schizophrenia, as only ~6 in 1000 cases carry a disruptive variant in one of those genes compared with ~1 in 1000 controls, a 6-fold enrichment in cases (similar to that we observe for NDD alleles in the present study). The SCHEMA study explicitly states that they have identified only a small fraction of the genes in which exonic variants contribute to the disorder (Singh et al. 2020); this is not to disparage the SCHEMA findings, which is a landmark paper on the (long) way forward in rare variant discovery, rather it is to stress the limitations. Given that, as well as key differences in the study designs (see next comment), we think it is not at all surprising that there is not a major overlap between the genes highlighted in that study and the ones we highlight here. Perhaps it is more of a surprise that there is any overlap at all (at *GRIA3*) given the low power of both studies.

9. Why did the authors only analyze *de novo* variants? Technically the same can be done on ultra-rare variants, which provides a greater power. Indeed, the large-scale ultra-rare variant study (these authors also coauthored) found that when focusing on the SCZ genes, "the majority of schizophrenia associations reported here appear to have little or no role in DD/ID despite the enormous power of published DD/ID studies to date". This further suggests that the *de novo* variants reported here are not the primary factors for SCZ.

We focussed on *de novo* variants for the following reasons. First, *de novo* variants are enriched for deleterious pathogenic mutations compared with inherited variants (Veltman and Brunner 2012). The expectation of a greater proportion of deleterious variants is equivalent to an expected improvement in the signal to noise, and therefore better power (for an equivalent sample size). This expectation is why so many studies of NDD disorders have focussed on the *de novo* paradigm, an expectation that has met with considerable success. It has also been empirically tested in the ASD paper of Satterstrom (2020), in which robust enrichments were seen for *de novo* mutations in ASD probands in expected gene sets (e.g mutation intolerant genes sets) but in exactly the same sample, there were no enrichments for transmitted alleles (Satterstrom et al. 2020). The second reason is that the NDD data we use to inform our analyses are exclusively taken from *de novo* variant studies. By

studying *de novo* variants in schizophrenia, we are performing an exact like for like test, which may be important given that the characteristics of *de novo* mutations and rare transmitted alleles are not the same; for example, the ASD paper of Satterstrom (2020) reported a substantially higher proportion of constrained PTV and missense variants among *de novo* variants compared with inherited variants (e.g. PTVs ($pLi > 0.995$) make up 5.13% of *de novo* mutations and 0.11% of inherited mutations, while missense ($MPC > 2$) variants make up 4.96% of *de novo* mutations and 1% of inherited mutations) (Satterstrom et al. 2020). Designating alleles as protein-truncating is relatively straightforward, whereas predicting the pathogenic effects of missense mutations is more difficult (a point raised by reviewer 3 point 2). Therefore, the effect of studying *de novo* versus transmitted alleles may be most important when it comes to missense mutations.

As noted above, we are unsure what the reviewer means by the 'primary factors' for schizophrenia, but would like to point out that our study and that of SCHEMA are not contradictory. SCHEMA observed that several of the genes they identified as associated with schizophrenia show no evidence for *de novo* variant enrichment in NDDs, suggesting these genes are not pleiotropic for schizophrenia and NDDs. In contrast, what we show is that when genes are pleiotropic for schizophrenia and NDDs, the class of mutation is generally congruent in schizophrenia and those disorders, and moreover, that the specific mutations contribute to both disorders.

Reviewer #2 (Remarks to the Author):

We thank the reviewer for the positive comments, and in particular their observation that "this is a key manuscript for the field that solidifies many of the earlier assumptions".

1) I have only one comment. The presentation of the data used and the flow of analyses could be more reader friendly. The group uses terms like "NDD gene set", "PTV/missense gene sets", "Primary set", Primary variant set", "negative control set" etc. Mostly these refer to variants, sometimes to patient groups. The manuscript does explain these terms, but it takes quite some digging. Would maybe a flow chart that describes the components and steps of the study help the reader to grasp the message of the study? This should be considered.

This is a fair point. We have made the manuscript more reader friendly by adding text to the results section to describe the NDD variant sets and our rationale for testing them (page 10 paragraph 1) and made the terminology used to describe NDD gene sets and NDD variant sets more consistent throughout the manuscript. We have also added the following figure (Figure 1) to summarise our genic pleiotropy and allelic pleiotropy study designs.

a

b

Figure 1. (A) Study design for analysis of genic pleiotropy. Genes enriched for protein-truncating variants or missense variants in neurodevelopmental disorders with a P value < 2.5×10^{-6} were identified from the Deciphering Developmental Disorders study (Kaplanis et al 2020). For each NDD gene set, the observed and expected number of *de novo* variants in 3,444 schizophrenia trios was compared using a two-sided two-sample Poisson rate ratio test. See online methods for further details. **(B) Study design for analysis of allelic pleiotropy.** Neurodevelopmental disorder variants were defined as *de novo* mutations reported in the largest

sequencing studies of developmental disorders (Kaplanis et al 2020) and autism spectrum disorders (Satterstrom et al 2020). Neurodevelopmental disorder variants were stratified into a NDD primary variant set, defined as those alleles annotated as protein-truncating variants affecting loss-of-function intolerant genes or missense variants with an MPC score ≥ 2 . The remaining neurodevelopmental disorder variants (protein-truncating variants not affecting loss-of-function intolerant genes, missense variants with an MPC score ≤ 2 and synonymous variants) were grouped into a NDD comparator variant set. The observed and expected number of schizophrenia *de novo* variants overlapping alleles within each NDD variant set was compared using a two-tailed Poisson exact test. See online methods for further details.

2) This reviewer would prefer that the Abstract would also have some actual numeric results, not just the story of the study.

We have added to the abstract the size of the schizophrenia and NDD samples analysed in our study and the P value for NDD variant enrichment.

Reviewer #3 (Remarks to the Author)

1) I thought early on the paper says that it presents new NDD cohorts. I don't see those reported in the Results. Indeed, I was expecting to see some Intro/Overview in the Results, showing the number of trio samples, the number of denovos, in new and published studies, and perhaps some clinical/phenotypic information about the cohorts.

All *de novo* data analysed in our study were derived from published schizophrenia and NDD data sets. In the introduction, to highlight the advance between this work and a previous study of ours, we wrote that we analysed sequencing data from "new large NDD cohorts" and we can see this was misleading. We thank the reviewer for drawing the ambiguous text to our attention, and we have updated the introduction to include the following text (page 5 paragraph 2).

"In the current study, we can now address these questions with the availability of the large and recently published NDD *de novo* variant data set from the Deciphering Developmental Disorders study (Kaplanis et al. 2020)."

As requested, we provide the number of schizophrenia trios, and *de novo* variants analysed, at the beginning of the results section (Page 7, paragraph 2).

"3,438 nonsynonymous *de novo* variants from 3,444 schizophrenia proband-parent trios (2,121 male and 1,323 female probands) were obtained from 11 published studies (see online methods for clinical information about these cohorts)."

We provide clinical information about the schizophrenia cohorts in the online methods, and note this in the main text on Page 7 paragraph 2:

2) There may be some problems with the prediction of variant category specific enrichments in both disorders, depending on the gene. First, effect sizes are stronger in NDD than SCZ, and for LOFs than for missense in both

disorders; therefore, there may be limited power to detect enrichments in SCZ missense denovos in Table 1 (indeed all the rate ratios are greater than 1 and likely not significantly different from those in LOF NDD genes vs missense NDD genes).

We agree that the contribution of *de novo* variants to NDDs is greater than it is to schizophrenia and that this contributes to the greater power of NDD studies. However, this does not change the interpretation of our results. Moreover, our study benefits from the substantial power of the NDD study as this increases the power of our own study by providing a large target set of NDD genes for us to test for pleiotropic effects.

We also agree that power to detect PTV enrichment is greater than power to detect missense enrichments. This means that where we see enrichments for the PTVs but not missense mutations, it is reasonable to ask if this simply reflects the relative power of the tests. We asked ourselves the same question, and this is one of the reasons why we conducted a direct test of the effect sizes of PTV versus missense enrichments. Our finding that this test is significant implies that, regardless of any differences in power, our study can estimate the effect size of the missense enrichment with enough precision to demonstrate that the two effect sizes are different.

In the second and third sections of Table 1, for each of the NDD gene sets defined fully or in part by missense mutations in NDD, we see missense mutations are significantly enriched in schizophrenia. Again, this is consistent with adequate power. Formal power calculations are impossible given the number of unknowns with respect to the shared genetic architecture of these disorders, but our findings suggest power to detect missense enrichments is not a key limitation to the study.

To further examine the hypothesis that pleiotropic genes are enriched for congruent classes of mutation across schizophrenia and NDDs, we evaluated at a genome-wide level whether per-gene NDD test statistics for PTV and missense variants can predict schizophrenia *de novo* variant enrichment (as suggested by the reviewer in the next point). This analysis showed that only NDD missense test statistics significantly predict schizophrenia missense enrichment (see Response Table R2 in the next comment), and confirms our view that the conclusions we derive from our study are not a consequence of inadequate power to detect missense enrichments.

Second, some proportion of predicted damaging missense variants are essentially LOFs; therefore, LOF and missense categories should show correlated effect sizes, which would make the predicted pattern less likely to be observed. I am not sure how to take these things into account in the analysis presented in Table 1; perhaps in the test statistics regression format, both LOF and missense NDD test statistics could be added to the model to assess whether both are associated with eg missense SCZ test statistics.

We agree that the unknown (but presumably non-negligible) proportion of missense mutations that are effectively LOF would have similar functional effects on a gene as PTVs, and that this could result in correlation among test statistics for PTV and missense variants. The effect of this would be to reduce power to see mutation

congruence so it would be more of a caveat if we had failed to see evidence for congruence.

While it is not possible to fully account for this possible correlation in the gene-set analysis approach presented in Table 1, we did try to mitigate this issue by generating 3 rather than 2 sets of genes; those where in NDDs, significant associations were found 1) for PTVs only 2) for missense mutations only and 3) for both classes of mutation. The purpose of that third set was to try to capture the set of genes where missense mutations tended to be LOF.

To more fully address the reviewer’s point, we have extended the multivariable Poisson regression model to evaluate the independent effects of NDD PTV and missense association test statistics on schizophrenia *de novo* variant enrichment at a genome wide level. The results show that NDD per-gene enrichment statistics significantly predict schizophrenia *de novo* variant enrichment, but only for congruent mutation classes (Response Table R2).

SZ DNV enrichment (dependent variable)	Genes tested	Predictors	Estimate	Std. Error	P
SZ PTVs	13015	NDD PTV P value	0.15	0.023	3.30E-11
		NDD miss P value	-0.018	0.041	0.67
SZ Missense	13015	NDD PTV P value	0.012	0.014	0.38
		NDD miss P value	0.049	0.017	0.0047

Response Table R2. Genome-wide analysis of mutation congruence in pleiotropic genes. A multivariable Poisson regression model was used to evaluate whether individual gene enrichment statistics in neurodevelopmental disorders predicts *de novo* variant enrichment in schizophrenia. This analysis tested 13,015 genes for which independent PTV and missense enrichment statistics were reported in the Deciphering Development Disorders study (Kaplanis et al 2020). SZ = schizophrenia; NDD PTV P value = per-gene enrichment P value for PTVs in developmental disorders; NDD miss P value = per-gene enrichment P value for missense variants in developmental disorders.

These findings provide further support for our conclusions drawn from our gene-set analysis and we have now included this analysis in the results section (page 9 paragraph 2) and added the above table to the supplementary material (Supplementary Table S1).

“To further examine the hypothesis that pleiotropic genes are enriched for congruent classes of mutation across schizophrenia and NDDs, we evaluated at a genome-wide level whether per-gene NDD test statistics for PTV and missense variants predict schizophrenia *de novo* variant enrichment. In the Deciphering Developmental Disorders study, independent PTV and missense enrichment statistics were both reported for 13,015 genes (missense P values are not available for 6,549 genes) (Kaplanis et al. 2020). These data were used in a multivariable Poisson regression model to evaluate the independent effects of NDD PTV and missense gene-wide test statistics on schizophrenia *de novo* variant enrichment. PTV enrichment in

schizophrenia was associated with the PTV enrichment test statistics in NDDs (beta = 0.15; P = 3.3×10^{-11}) but not with the NDD missense enrichment test statistics (beta = -0.018; P = 0.67; Supplementary Table S1). Conversely, missense enrichment in schizophrenia was associated with the NDD missense enrichment test statistics (beta = 0.049; P = 0.0047) but not with the NDD PTV enrichment test statistics (beta = 0.012; P = 0.38; Supplementary Table S1)."

4) The test statistic regression framework is potentially a nice one for addressing this kind of question. Genes' pLI status, or eg brain expression, could be added as covariates to test whether they impact the pleiotropy. However, it may be possible to observe test statistic correlations due to mutation rate (/gene length), in the same way we see correlations among rare gene burden test statistics which we attribute to the same genes being underpowered across disorders, so we might filter for genes with some minimum number of counts (perhaps some minimum mutation rate for denovo analyses). Perhaps some simulations could be used to show whether this may be an issue.

These are very good suggestions which we have now implemented. As suggested, we have adapted the Poisson regression framework to covary for per gene measures of brain expression and purifying selection. Measures of gene expression in brain tissue were obtained from the BrainSeq project and defined as the level of expression in dorsolateral prefrontal cortex averaged over all available timepoints (second trimester to 85 years) (Jaffe et al. 2018). For per gene measures of purifying selection, we used the gnomAD observed/expected constraint score (Karczewski et al. 2020), which is the ratio of observed / expected number of variants per gene for a given mutation class. The results reported in the main text (Supplementary Table S1 and Response Table S2) do not change after covarying for brain expression and purifying selection (see Response Table R3 below). Thus, brain expression and gene constraint scores do not impact our findings.

We have now added the following text to the results section (page 9 paragraph 2).

"Our finding that genes that are pleiotropic for schizophrenia and NDDs tend to be enriched for congruent classes of variant across these disorders was robust to inclusion of measures of gene brain expression and gene constraint as covariates to the multivariable Poisson regression model (see Supplementary Table S2 and online methods for further details). These results extend the findings based only on genes that met exome-wide significance in NDDs (Table 1), and suggest that congruent variant classes within pleiotropic genes contribute to risk of schizophrenia and NDDs."

We have also added Response Table R3 to the supplementary material (Supplementary Table S2).

SZ DNV enrichment (Dependent variable)	Genes tested	Predictors	Estimate	Std. Error	P
SZ PTVs	12893	NDD PTV P value	0.14	0.023	1.20E-09
		NDD Miss P value	-0.034	0.042	0.41
		Brain_exp	0.069	0.035	0.053
		PTV o/e	-0.36	0.22	0.11
SZ Missense	12991	NDD PTV P value	0.012	0.014	0.38
		NDD Miss P value	0.048	0.018	0.0069
		Brain_exp	0.021	0.014	0.14
		Miss o/e	0.11	0.13	0.41

Response Table R3. Genome-wide analysis of mutation congruence in pleiotropic genes including gene brain expression and gene constraint scores as covariates. Schizophrenia *de novo* variant enrichments for each mutation class are predicted by NDD gene P values for the same mutation class after covarying for both gene expression and constraint. SZ = schizophrenia; NDD PTV P value = per-gene enrichment P value for PTVs in developmental disorders; NDD miss P value = per-gene enrichment P value for missense variants in developmental disorders. PTV obs/exp = observed / expected number of protein-truncating variants per-gene in the gnomAD database. Miss obs/exp = observed / expected number of missense variants per-gene in the gnomAD database.

We have done also, as the reviewer suggests, repeated our analysis after excluding genes with the lowest mutation rates. Here, we first binned per gene mutation rates into deciles, and then sequentially excluded genes from lower deciles from our analyses. These results are consistent with those reported in our analysis of all genes (see Response Table R4 below). Given we provide empirical evidence that our results are not biased by low per gene mutation rates, we believe that simulations are not required to address the impact of mutation rates on our results.

SZ DNV enrichment (Dependent variable)	Mut. rate bottom deciles removed	N genes tested	Predictors	Estimate	Std. Error	P
PTV	0 (original analysis)	13015	NDD PTV P value NDD miss P value	0.15 -0.018	0.023 0.041	3.30E-11 0.67
	1	12246	NDD PTV P value NDD miss P value	0.15 -0.015	0.024 0.042	2.70E-10 0.72
	2	11296	NDD PTV P value NDD miss P value	0.16 -0.022	0.025 0.044	1.30E-10 0.62
	3	10225	NDD PTV P value NDD miss P value	0.17 -0.024	0.027 0.046	1.10E-10 0.6
	4	9074	NDD PTV P value NDD miss P value	0.18 -0.017	0.029 0.048	1.30E-09 0.72
	5	7804	NDD PTV P value NDD miss P value	0.2 -0.045	0.03 0.052	6.10E-11 0.4
Missense	0 (original analysis)	13015	NDD PTV P value NDD miss P value	0.012 0.049	0.014 0.017	0.38 0.0047
	1	12417	NDD PTV P value NDD miss P value	0.014 0.047	0.015 0.018	0.33 0.0085
	2	11565	NDD PTV P value NDD miss P value	0.015 0.048	0.016 0.018	0.34 0.0079
	3	10519	NDD PTV P value NDD miss P value	0.015 0.047	0.017 0.019	0.36 0.013
	4	9343	NDD PTV P value NDD miss P value	0.015 0.047	0.018 0.02	0.39 0.018
	5	8067	NDD PTV P value NDD miss P value	0.016 0.046	0.019 0.021	0.4 0.025

Response Table R4. Genome-wide analysis of mutation congruence in pleiotropic genes after excluding genes with the lowest mutation rates. The top rows within the “SZ DNV enrichment” PTV and missense sections show results from our original analysis of all genes (i.e. Response Table R2). After removing genes with the lowest mutation rates (divided into deciles 1 to 5), the results from this multivariable Poisson regression are consistent with those reported in our original analysis.

5) SUGGESTIONS (perhaps some of these are in the manuscript, but could be emphasized more, or have analyses done and move from Discussion to Results):

Given the timeliness mentioned above, that the SCHEMA paper is forthcoming and (on the preprint server and in recent meetings) presents at least one example of differential mutation categories between the disorders, it might be nice to drill down on Table 1, to show which genes seem to drive the congruence and how many and which genes may not. Are the 20 LOFs and 6 missense SCZ denovos scattered randomly across genes or concentrated in one or a few genes?

We have now expanded Supplementary Table S12, which originally displayed the NDD genes included in our gene set analysis, to also show the number of schizophrenia missense and PTV *de novo* variants observed in each NDD gene. For the 283 NDD genes tested, 35 were disrupted by a single *de novo* variant in schizophrenia, and 8 and 3 genes were disrupted by 2 and 3 *de novo* variants in schizophrenia, respectively. No NDD gene was disrupted by > 3 schizophrenia variants. Therefore, the evidence for genic pleiotropy reported in our paper comes from mutations scattered across many genes.

6) Subphenotypes relating to cognition/school performance in SCZ probands. I would be interested in more description of this for the NDD/ASD probands (suggesting to expand the last Results subsection). Do NDD probands with denovos in SCZ denovo genes have any particular symptoms relating to cognition or psychosis? Are they more or less likely to have multi-system syndromic symptomology than NDD probands with denovos in non-SCZ denovo genes? (Is it possible to assess the frequency of different symptoms across OMIM phenotypes, for example?) Could you present some tables of the frequencies of these things, if not statistical test(s), in addition to the anecdotal list and supp table?

We agree these are interesting questions. We have been able to obtain IQ scores and age at walking for the ASD probands who carry the same mutation as people with schizophrenia. However, we do not have access to the relevant phenotypic data from the NDD and ASD cohorts to address the other questions. The following text has now been added (page 14 paragraph 3).

"Three ASD probands carried a NDD primary variant that was observed in our schizophrenia sample. Two ASD probands shared the same *SCN2A* missense variant (Table 2), both of whom have comorbid intellectual disability (IQ scores of 50 and 69) (Satterstrom et al. 2020). The schizophrenia proband who had this *SCN2A* missense variant developed psychotic symptoms at an early age of 14 years, but does not have a learning disability or comorbid ASD (Supplementary Table S8). The other ASD proband shared a missense variant in *CSNK2A1* with a schizophrenia proband (Table 2). This ASD proband did not have intellectual disability (IQ of 92), but showed delayed age of walking (Satterstrom et al. 2020)"

We have also added a new column to Supplementary Table S10 to show recurrent phenotypes observed in people from the 2017 Deciphering Developmental Disorders study who carried any *de novo* missense or PTV mutation within *CSNK2A1*, *SCN2A*, *AUTS2* and *SLC6A1*.

MINOR COMMENTS/QUESTIONS:

7) Table 2, are there really genes with Exac pLi ~ 10⁻¹⁶? I thought those were rank percentiles across genes (ie minimum 1/20,000). Just pointing it out in case of typo.

We can confirm that the pLi scores we previously reported in Table 2 are correct. pLi is a per gene probability for being intolerant to loss of function mutations. pLi scores have a bimodal distribution, with most genes tolerant (scores close to 0) or intolerant

(scores close to 1). The lowest pLi score reported in gnomAD, that for *OBSCN*, is 1.86×10^{-164} . However, to simplify the presentation, we have now truncated Table 2 to only show variants observed in schizophrenia from our NDD primary variant set, and the pLi scores that the reviewer mentions are presented in Supplementary Table S3

8) When you look in SCZ case/control exomes, why not refer to the recent SCHEMA results (which are publicly available)?

The SCHEMA dataset represents an extremely complex meta-analysis of datasets that differ in many important ways, making simple allele count lookups misleading. Examples of important variables that have to be accounted for in the analysis include ancestral population, sequencing centre, and exome capture kit. The development of the analytic methodology to allow all of the above to be correctly adjusted for one of the main triumphs of the primary SCHEMA manuscript, but implementing this requires access to all the raw data. At present only summary data for SCHEMA are available, which is reasonable given the primary paper itself is still under review.

References

- Brown, Lawrence D., T. Tony Cai, and Anirban DasGupta. 2002. 'Confidence Intervals for a Binomial Proportion and Asymptotic Expansions'. *The Annals of Statistics* 30 (1). <https://doi.org/10.1214/aos/1015362189>.
- Chiu, A. T. G., S. L. C. Pei, C. C. Y. Mak, G. K. C. Leung, M. H. C. Yu, S. L. Lee, M. Vreeburg, et al. 2018. 'Okur-Chung Neurodevelopmental Syndrome: Eight Additional Cases with Implications on Phenotype and Genotype Expansion'. *Clinical Genetics* 93 (4): 880–90. <https://doi.org/10.1111/cge.13196>.
- Howrigan, Daniel P., Samuel A. Rose, Kaitlin E. Samocha, Menachem Fromer, Felecia Cerrato, Wei J. Chen, Claire Churchhouse, et al. 2020. 'Exome Sequencing in Schizophrenia-Affected Parent-Offspring Trios Reveals Risk Conferred by Protein-Coding de Novo Mutations'. *Nature Neuroscience* 23 (2): 185–93. <https://doi.org/10.1038/s41593-019-0564-3>.
- Jaffe, Andrew E., Richard E. Straub, Joo Heon Shin, Ran Tao, Yuan Gao, Leonardo Collado-Torres, Tony Kam-Thong, et al. 2018. 'Developmental and Genetic Regulation of the Human Cortex Transcriptome Illuminate Schizophrenia Pathogenesis'. *Nature Neuroscience* 21 (8): 1117–25. <https://doi.org/10.1038/s41593-018-0197-y>.
- Kaplanis, Joanna, Kaitlin E. Samocha, Laurens Wiel, Zhancheng Zhang, Kevin J. Arvai, Ruth Y. Eberhardt, Giuseppe Gallone, et al. 2020. 'Evidence for 28 Genetic Disorders Discovered by Combining Healthcare and Research Data'. *Nature*, October, 1–7. <https://doi.org/10.1038/s41586-020-2832-5>.
- Karczewski, Konrad J., Laurent C. Francioli, Grace Tiao, Beryl B. Cummings, Jessica Alföldi, Qingbo Wang, Ryan L. Collins, et al. 2020. 'The Mutational Constraint Spectrum Quantified from Variation in 141,456 Humans'. *Nature* 581 (7809): 434–43. <https://doi.org/10.1038/s41586-020-2308-7>.
- Niemi, Mari E. K., Hilary C. Martin, Daniel L. Rice, Giuseppe Gallone, Scott Gordon, Martin Kelemen, Kerrie McAloney, et al. 2018. 'Common Genetic Variants Contribute to Risk of Rare Severe Neurodevelopmental Disorders'. *Nature* 562 (7726): 268–71. <https://doi.org/10.1038/s41586-018-0566-4>.

- Rees, Elliott, Jun Han, Joanne Morgan, Noa Carrera, Valentina Escott-Price, Andrew J. Pocklington, Madeleine Duffield, et al. 2020. 'De Novo Mutations Identified by Exome Sequencing Implicate Rare Missense Variants in SLC6A1 in Schizophrenia'. *Nature Neuroscience* 23 (2): 179–84. <https://doi.org/10.1038/s41593-019-0565-2>.
- Satterstrom, F. Kyle, Jack A. Kosmicki, Jiebiao Wang, Michael S. Breen, Silvia De Rubeis, Joon-Yong An, Minshi Peng, et al. 2020. 'Large-Scale Exome Sequencing Study Implicates Both Developmental and Functional Changes in the Neurobiology of Autism'. *Cell* 180 (3): 568-584.e23. <https://doi.org/10.1016/j.cell.2019.12.036>.
- Singh, Tarjinder, Mitja I. Kurki, David Curtis, Shaun M. Purcell, Lucy Crooks, Jeremy McRae, Jaana Suvisaari, et al. 2016. 'Rare Loss-of-Function Variants in SETD1A Are Associated with Schizophrenia and Developmental Disorders'. *Nat Neurosci* 19 (4): 571–77. <https://doi.org/10.1038/nn.4267>.
- Singh, Tarjinder, Timothy Poterba, David Curtis, Huda Akil, Mariam Al Eissa, Jack D. Barchas, Nicholas Bass, et al. 2020. 'Exome Sequencing Identifies Rare Coding Variants in 10 Genes Which Confer Substantial Risk for Schizophrenia'. *MedRxiv*, September, 2020.09.18.20192815. <https://doi.org/10.1101/2020.09.18.20192815>.
- Tansey, K. E., E. Rees, D. E. Linden, S. Ripke, K. D. Chambert, J. L. Moran, S. A. McCarroll, et al. 2016. 'Common Alleles Contribute to Schizophrenia in CNV Carriers'. *Molecular Psychiatry* 21 (8): 1085–89. <https://doi.org/10.1038/mp.2015.143>.
- Veltman, Joris A., and Han G. Brunner. 2012. 'De Novo Mutations in Human Genetic Disease'. *Nat Rev Genet* 13 (8): 565–75. <https://doi.org/10.1038/nrg3241>.
- Weiner, Daniel J., Emilie M. Wigdor, Stephan Ripke, Raymond K. Walters, Jack A. Kosmicki, Jakob Grove, Kaitlin E. Samocha, et al. 2017. 'Polygenic Transmission Disequilibrium Confirms That Common and Rare Variation Act Additively to Create Risk for Autism Spectrum Disorders'. *Nat Genet* 49: 978. <https://doi.org/10.1038/ng.3863>.

Reviewers' Comments:

Reviewer #1:

Remarks to the Author:

I appreciate the authors' detailed response. They have addressed most of my comments, but I would like to follow up on two points:

1. In my original comments #3 and #6, I was trying to assess if NDD should be considered as a comorbidity in a subset of schizophrenia patients (and vice versa). For example, schizophrenia patients with pathogenic CNVs or SNVs are known to have lower IQ, but they are not considered to have NDD unless their IQs are lower than a cutoff for intellectual disability. Is it possible that there is mild NDD phenotype (such as low IQ) in schizophrenia patients which is missed by diagnosis? Similarly, is it possible that schizophrenia is not (yet) assessed in NDD patients? As the authors suggested, "it would still imply that the mutations can manifest as both NDD and schizophrenia", but wouldn't it be more appropriate to call them syndromic form of schizophrenia?

2. In relevant to the point above, have they assessed the IQ of the schizophrenia patients with de novo variant in NDD genes? Are they different from other patients? This was supposed to be my original comment #5 that was missing.

Reviewer #3:

Remarks to the Author:

The authors have made a comprehensive effort to address the comments of all the reviewers, and I hope it has led to some improvements to the paper (given the added work and time!). I am satisfied as to the validity of results presented and the importance/potential impact of this paper. Signed, Eli Stahl.

- 1. In my original comments #3 and #6, I was trying to assess if NDD should be considered as a comorbidity in a subset of schizophrenia patients (and vice versa). For example, schizophrenia patients with pathogenic CNVs or SNVs are known to have lower IQ, but they are not considered to have NDD unless their IQs are lower than a cutoff for intellectual disability. Is it possible that there is mild NDD phenotype (such as low IQ) in schizophrenia patients which is missed by diagnosis? Similarly, is it possible that schizophrenia is not (yet) assessed in NDD patients? As the authors suggested, “it would still imply that the mutations can manifest as both NDD and schizophrenia”, but wouldn't it be more appropriate to call them syndromic form of schizophrenia?**

We thank the reviewer for the opportunity to address these clinically important issues, to which we should probably have devoted more discussion in our paper.

The reviewer asks whether we should distinguish between syndromic and non-syndromic forms of schizophrenia based on IQ deficits and other features of NDDs. Currently, schizophrenia is a clinical diagnosis, based on information obtained from the case history and examination of the mental state, and there is no agreed way to confirm, reject, or subclassify the diagnosis based on pathology, laboratory measures or biomarkers of any kind, including cognitive tests or features suggestive of NDDs. The idea that we might be able to define a valid subtype or subtypes based on cognitive or other features is certainly of great interest and is one we have considered carefully. However, while we acknowledge that the situation may change as more data accrue, we do not believe that currently there are sufficient grounds for taking such a step. Rather, we believe that the evidence better supports the view that there is a spectrum of neurodevelopmental impairment in those with schizophrenia, rather than a dichotomy between disorders that represent “true schizophrenia” and disorders that can be considered pure manifestations of a genomic disorder or syndrome (Owen and O'Donovan 2017). For example, IQ is continuously distributed in schizophrenia, but with a mean reduction of approximate 1 standard deviation relative to the general population (Green, Horan, and Lee 2019). The evidence also suggests that IQ has a monotonic relationship with schizophrenia risk across the IQ range (Kendler et al. 2014) rather than there being an IQ threshold that is associated with a step change in liability. Moreover, stratification of individuals with schizophrenia by low IQ does not identify a group of individuals in whom common risk variants, identified by GWAS, do not contribute to liability, and lower cognitive ability generally in schizophrenia is not associated with lower burden of common alleles that confer risk to what might be considered general forms of the disorder (Richards et al. 2020). Finally, we have shown that in those in whom it is very tempting to assign schizophrenia to a syndromic cause, i.e., people with deletion of 22q11.2, schizophrenia is the result of both the CNV and common variant liability that is shared with general forms of the disorder (Cleyne et al. 2020). Thus, while we cannot exclude the idea that true syndromic causes of schizophrenia may be identified in the future, for now, we do not think a dichotomy based on cognitive ability or NDD features is justified.

The reviewer is correct that a diagnosis of co-morbid intellectual disability in schizophrenia is based on an arbitrary cut-off of IQ <70. They are also correct to point out that cognitive function is frequently not tested clinically in schizophrenia and that functionally impactful impairments (whether > or < 70 IQ points) are frequently missed. This argues that assessment of cognitive function should be an integral part of the clinical workup for schizophrenia. Indeed, this was initially proposed for DSM5 but moved from the main part of the manual late in the review process (Green, Horan, and Lee 2019). In contrast, in ICD-11 the level of cognitive impairment is listed as a qualifier for schizophrenia that should be rated along with other key features. However, for the reasons outlined above, we believe that the clinical utility of measuring cognitive function in schizophrenia will be to understand individual

impairments and identify the need for remediation rather than to define diagnostically distinct subgroups.

It is also true as the reviewer points out that there may be instances in which features of schizophrenia are missed in individuals diagnosed with NDDs, particularly in those with impaired communication skills. We hope that our paper will highlight the clinical need for greater awareness of the potential for comorbidity between schizophrenia and neurodevelopmental disorders.

In response to the reviewer's comment, we have now added the following text to the discussion.

Page 17 paragraph 2:

“These findings do not support the suggestion that carriers of NDD mutations necessarily present with clear syndromic features, based on cognitive function or NDD comorbidity. This is clearly relevant to the question of whether we should distinguish between syndromic and non-syndromic forms of schizophrenia based on IQ deficits and other features of NDDs. Currently, schizophrenia is a clinical diagnosis, using information obtained from the case history and examination of the mental state, and there is no agreed way to confirm, reject, or subclassify the diagnosis based on pathology, laboratory measures or biomarkers of any kind, including cognitive tests or features suggestive of NDDs. The idea that we might be able to define a valid subtype or subtypes based on cognitive or other features is certainly of great interest and is one we have considered carefully. However, while we acknowledge that the situation may change as more data accrue, we do not believe that there are sufficient grounds for taking such a step. Rather, we believe that the evidence better supports the view that there is a spectrum of neurodevelopmental impairment in those with schizophrenia, rather than a dichotomy between disorders that represent “true schizophrenia” and disorders that can be considered pure manifestations of a genomic disorder or syndrome (Owen and O'Donovan 2017). For example, IQ is continuously distributed in schizophrenia, but with a mean reduction of approximate 1 standard deviation relative to the general population (Green, Horan, and Lee 2019). The evidence also suggests that IQ has a monotonic relationship with schizophrenia risk across the IQ range (Kendler et al. 2014) rather than there being an IQ threshold that is associated with a step change in liability. Moreover, lower cognitive ability in schizophrenia is not associated with a lower burden of common alleles that confer risk to what might be considered general forms of schizophrenia (Richards et al. 2020). Finally, for individuals in whom it is very tempting to assign schizophrenia to a syndromic cause, i.e., people with deletion of 22q11.2, schizophrenia is the result of both the CNV and common variant liability that is shared with general forms of the disorder (Cleyen et al. 2020). Thus, while we cannot exclude the idea that true syndromic causes of schizophrenia may be identified in the future, for now, we do not think a dichotomy based on cognitive ability or NDD features is justified. This highlights the need for additional studies to systemically evaluate the effect of NDD variants on quantitative measures of cognitive function and other biomarkers in schizophrenia.”

Page 18 paragraph 2:

“It is possible for features of schizophrenia to sometimes be missed in individuals diagnosed with NDDs, particularly in those with impaired communication skills. It is also the case that cognitive function is frequently not tested clinically in schizophrenia and that functionally impactful impairments, whether or not they satisfy formal diagnostic criteria for ID, are frequently missed. This argues that assessment of cognitive function should be an integral part of the clinical workup for schizophrenia as recently suggested in ICD-11 (World Health Organization 2018). Thus, our findings highlight the clinical need for greater awareness of potential comorbidity between schizophrenia and neurodevelopmental disorders.”

2. In relevant to the point above, have they assessed the IQ of the schizophrenia patients with de novo variant in NDD genes? Are they different from other patients? This was supposed to be my original comment #5 that was missing.

IQ data are not available for almost all schizophrenia samples included in our study, and therefore we are unable to systematically evaluate differences in cognitive function between schizophrenia carriers of NDD variants and schizophrenia non-carriers. However, we provide indirect measures of cognitive function, along with any evidence for additional NDD phenotypes, in schizophrenia patients carrying specific NDD variants (Table S8 and main text). Thus on page 16 we point out that only 2 of the 9 probands with a *de novo* variant in the NDD primary variant set had evidence of clinically significant premorbid intellectual impairment, consistent with findings from other studies showing damaging *de novo* mutations may be enriched in, but are not confined to, those with intellectual disability (Fromer et al. 2014; Singh et al. 2017). These findings, together with the absence of evidence for pre-existing autism or related disorders (Supplementary Table S8), are of clinical relevance since they suggest that patients presenting with schizophrenia in the absence of neurodevelopmental comorbidities may carry damaging mutations that are associated with more severe neurodevelopmental outcomes. These findings do not support the suggestion that carriers of NDD mutations necessarily present with clear syndromic features, based on cognitive function or NDD comorbidity.

We agree that it will be important for future studies to systemically evaluate the effect of NDD variants on quantitative measures cognitive function in schizophrenia, and have added the following text to the discussion (page 17 paragraph 2).

“Thus, while we cannot exclude the idea that true syndromic causes of schizophrenia may be identified in the future, for now, we do not think a dichotomy based on cognitive ability or NDD features is justified. This highlights the need for additional studies to systematically evaluate the effect of NDD variants on quantitative measures of cognitive function and other biomarkers in schizophrenia.”

Response references

- Cleynen, Isabelle, Worrawat Engchuan, Matthew S. Hestand, Tracy Heung, Aaron M. Holleman, H. Richard Johnston, Thomas Monfeuga, et al. 2020. ‘Genetic Contributors to Risk of Schizophrenia in the Presence of a 22q11.2 Deletion’. *Molecular Psychiatry*, February. <https://doi.org/10.1038/s41380-020-0654-3>.
- Fromer, Menachem, Andrew J Pocklington, David H Kavanagh, Hywel J Williams, Sarah Dwyer, Padhraig Gormley, Lyudmila Georgieva, et al. 2014. ‘De Novo Mutations in Schizophrenia Implicate Synaptic Networks’. *Nature* 506: 179–84.
- Green, Michael F., William P. Horan, and Junghee Lee. 2019. ‘Nonsocial and Social Cognition in Schizophrenia: Current Evidence and Future Directions’. *World Psychiatry* 18 (2): 146–61. <https://doi.org/10.1002/wps.20624>.
- Kendler, Kenneth S., Henrik Ohlsson, Jan Sundquist, and Kristina Sundquist. 2014. ‘IQ and Schizophrenia in a Swedish National Sample: Their Causal Relationship and the Interaction of IQ With Genetic Risk’. *American Journal of Psychiatry* 172 (3): 259–65. <https://doi.org/10.1176/appi.ajp.2014.14040516>.
- Owen, Michael J., and Michael C. O’Donovan. 2017. ‘Schizophrenia and the Neurodevelopmental Continuum: Evidence from Genomics’. *World Psychiatry* 16 (3): 227–35. <https://doi.org/10.1002/wps.20440>.
- Richards, Alexander L., Antonio F. Pardiñas, Aura Frizzati, Katherine E. Tansey, Amy J. Lynham, Peter Holmans, Sophie E. Legge, et al. 2020. ‘The Relationship Between

- Polygenic Risk Scores and Cognition in Schizophrenia'. *Schizophrenia Bulletin* 46 (2): 336–44. <https://doi.org/10.1093/schbul/sbz061>.
- Singh, Tarjinder, James T. R. Walters, Mandy Johnstone, David Curtis, Jaana Suvisaari, Minna Torniainen, Elliott Rees, et al. 2017. 'The Contribution of Rare Variants to Risk of Schizophrenia in Individuals with and without Intellectual Disability'. *Nat Genet* 49 (8): 1167–73. <https://doi.org/10.1038/ng.3903>.
- World Health Organization. 2018. 'International Classification of Diseases for Mortality and Morbidity Statistics (11th Revision).', Retrieved from <https://icd.who.int/browse11/l-m/en>.

Reviewers' Comments:

Reviewer #1:

Remarks to the Author:

The authors have adequately addressed my concerns. I have no additional comments.